# Interpretation of Fluoride Groundwater Contamination in Tamnar Area, Raigarh, Chhattisgarh, India

Mirza Kaleem Beg [1],*, Navneet Kumar [2,3], S. K. Srivastava [4] and E. J. M. Carranza [5]

1   Chhattisgarh Space Applications Centre, CCOST, Raipur 492001, India
2   Center for Development Research (ZEF), University of Bonn, 53113 Bonn, Germany; nkumar@uni-bonn.de
3   Global Mountain Safeguard Research (GLOMOS), United Nations University, UN Campus, Platz der Vereinten Nationen 1, 53113 Bonn, Germany
4   Indian Institute of Remote Sensing, Dehradun 248001, India
5   Department of Geology, University of the Free State, Bloemfontein 9301, South Africa
*   Correspondence: mkbaig77@gmail.com

**Abstract:** A high concentration of fluoride ($F^-$) in drinking water is harmful and is a serious concern worldwide due to its toxicity and accumulation in the human body. There are various sources of fluoride ($F^-$) and divergent pathways to enter into groundwater sources. High $F^-$ incidence in groundwater was reported in Raigarh district of Central India in a sedimentary (Gondwana) aquifer system. The present study investigates the hydrogeochemistry of groundwater in the Tamnar area of Raigarh district to understand the plausible cause(s) of high $F^-$ concentration, especially the source(s) and underlying geochemical processes. Groundwater samples, representing pre-monsoon (N = 83), monsoon (N = 20), and post-monsoon (N = 81) seasons, and rock samples (N = 4) were collected and analyzed. The study revealed that (i) groundwater with high $F^-$ concentration occurs in the Barakar Formation, which has a litho-assemblage of feldspathic sandstones, shales, and coal, (ii) high $F^-$ concentration is mainly associated with Na-Ca-HCO$_3$, Na-Ca-Mg-HCO$_3$, and Na-Mg-Ca-HCO$_3$ types of groundwater, (iii) the $F^-$ concentration increases as the ratio of Na$^+$ and Ca$^{2+}$ increases (Na$^+$: Ca$^{2+}$, concentration in meq/l), (iv) $F^-$ has significant positive correlation with Na$^+$ and SiO$_2$, and significant negative correlation with Ca$^{2+}$, Mg$^{2+}$, HCO$_3^-$, and TH, and (v) high $F^-$ concentration in groundwater is found in deeper wells. Micas and clay minerals, occurring in the feldspathic sandstones and intercalated shale/clay/coal beds, possibly form an additional source for releasing $F^-$ in groundwater. Feldspar dissolution coupled with anion (OH$^-$ or F$^-$) and cation (Ca$^{2+}$ for Na$^+$) exchange are probably the dominant geochemical processes taking place in the study area. The higher residence time and temperature of groundwater in deeper aquifers also play a role in enhancing the dissolution of fluorine-bearing minerals. Systematic hydrogeochemical investigations are recommended in the surrounding area having a similar geologic setting in view of the potential health risk to a large population.

**Keywords:** geochemical process; Barakar Formation; hydrogeochemistry; groundwater contamination

## 1. Introduction

The sustainability and quality of groundwater are crucial, as it is relied upon by approximately 2.5 billion people globally [1]. However, natural and human-related factors cause fluctuations in the availability and quality of groundwater [2,3]. The quality of groundwater has a significant role in the prevalence of various diseases. Among the various water quality parameters, fluoride is a significant contaminant that poses serious health risks and has garnered attention [4]. Fluoride ($F^-$) in groundwater can be beneficial for human health within a specific range; however, when the concentration is too low or high, it can have adverse effects. Low levels of fluoride in groundwater can lead to dental caries and poor bone development, while excessive intake can cause dental fluorosis and harm to the kidneys, bones, reproductive organs, nerves, and muscles [2]. The optimum

range of $F^-$ in drinking water can differ as a function of various factors such as environmental conditions and socio-economic factors. The World Health Organization (WHO) has specified that the safe permissible limit of $F^-$ in drinking water is 1.5 mg/L [5]. The primary source of intake of $F^-$ by humans is groundwater, as it is the primary drinking water source in rural and urban areas [6]. The natural occurrence of high fluoride concentrations in groundwater is a global health concern that is potentially affecting hundreds of millions of people, predominantly in the Global South [7]. A large number of people in 67 countries, including India, suffer from endemic fluorosis due to excess $F^-$ content in groundwater [8–12]. In India, the $F^-$ problem in groundwater is reported to occur in varied geological and environmental settings [13]. In 1937, high $F^-$ problem was first observed in Nellore District of Andhra Pradesh of India. According to UNICEF, at least 177 districts in 19 States of India were affected by excess $F^-$ levels in groundwater [14,15]. It was reported that around 62 million people from these States are largely dependent on groundwater and suffer from fluorosis [13,16]. Further, about 60–70 million people were estimated to be at risk [17,18].

Geogenic origin, i.e., weathering, dissolution, and leaching of fluoride-bearing minerals into underground water through bedrock, is considered the major source of $F^-$ in groundwater [19–22]. In addition to the geogenic factors, $F^-$ contamination in groundwater takes place from anthropogenic sources, viz. agriculture industries using extensively high amounts of phosphatic fertilizers, industries using coal for thermal power, and discharges from industries [10,19,23,24]. Mining activities and heavy groundwater exploitations enhance the dissolution rate of fluoride [25]. The mineral and chemical composition of bedrock is, therefore, considered one of the major factors contributing to the occurrence of $F^-$ in groundwater. The natural path of $F^-$ enrichment in groundwater depends largely on several factors, such as rock chemistry, solubility of fluoride-bearing minerals, temperature, pH, bedrock constituting the aquifers and their anion-exchange capacities, residence time of water in aquifers or duration of rock–water interaction, climate, well depth, and geological structures [18,21,26,27]. Numerous studies have been conducted in different geologic settings by researchers across the world to understand the linkage of $F^-$ occurrence in groundwater vis-à-vis rock–water interaction and the geochemical processes. Earlier studies revealed that $F^-$ has a higher affinity for sodium than calcium; hence, the sodium bicarbonate ($NaHCO_3$) type of water decreases with calcium ions and increases with sodium ions, and it has a neutral-to-alkaline pH, indicating favorability of chemical conditions for fluoride dissolution processes that accelerate $F^-$ concentration in subsurface water [28–30]. It was found that groundwater with elevated levels of $F^-$ is generally characterized by (a) high $HCO_3$ alkalinity, $Na^+$, pH, and silica, and (b) low $Ca^{2+}$ and hardness [18,31–33]. Generally, a negative correlation between $F^-$ and $Ca^{2+}$ in groundwater, including Indian groundwater, has been observed by several researchers (WHO, 2017). Cation/base-exchange ($Ca^{2+}$ for $Na^+$) and anion-exchange ($OH^-$ for $F^-$) geochemical processes also promote an increase in $F^-$ levels in groundwater [31,33–35]. Deeper wells are generally found to contain higher $F^-$ concentrations as compared to shallow wells because of the increased solubility of minerals with an increase in temperature [5,36]. Further, groundwater in arid areas has comparatively higher $F^-$ concentration than in humid areas [37].

This study constitutes a hydrogeochemical investigation in a sedimentary formation of Gondwana Supergroup coal-bearing rocks located in central India, an area known for potential coal-mining activities. The harmful effects of $F^-$ incidence in groundwater on the people's health such as several incidences of dental and skeletal fluorosis have been reported in the study area [38]. Given the large populations at risk due to high fluoride concentration in drinking water sources, there is an urgent need for a systematic and scientific investigation of $F^-$ occurrence in groundwater [39]. This study aims to analyze the high fluoride concentration in groundwater during pre-, post-, and mid-monsoon periods in the Barakar Formation, a particular geological formation, to understand fluoride

enrichment in groundwater and geology, and to determine geochemical behavior in deep bore wells by conducting hydrogeochemical analysis.

## 2. Materials and Methods

### 2.1. Study Area

The study area forms part of the Pahaj River watershed in Tamnar Block of Raigarh District, Chhattisgarh State, India. It covers 240 km$^2$ and is bound by latitudes 22°05′00″ N–22°15′00″ N and longitudes 83°20′00″ E–83°30′00″ E. Tamnar town, the Block headquarters, is located near the southern boundary of the study area. The Pahaj River, a fourth-order stream and tributary of the Kelo River, flows from north to south and divides the study area into two nearly equal parts (Figure 1). The elevation in the area ranges from about 260 to 580 m above mean sea level (amsl). While low hills and ridges occur in the NE and SW parts, most of the area is characterized by nearly flat to gently sloping pediplain surfaces with occasional outcrops. The typical average annual rainfall in the district is about 1580 mm, with >85% of rainfall taking place during the Indian summer monsoon (rainy) season. The onset of monsoon season takes place in mid-June and lasts until September. The minimum and maximum temperatures (monthly means) vary from ~10 °C in January to ~47 °C in May. Groundwater forms the source of drinking water in the area. The study area is dominated by the coal-bearing rocks of the Gondwana Supergroup (Lower Gondwana, age ranging from Lower Permian to Early Triassic), constituting a thick sequence of alternating sandstones, shales, clays, and coal beds (Table 1, Figure 1). The general trend of the rock formations is NW–SE, and dips are generally <5°. The Gondwana rocks that form the aquifers have both primary and secondary porosity/permeability; the latter is imparted by weathering and fracturing. The alternation of sandstones, shales, clays, and coal beds typically give rise to multi-tier aquifer system, sandstones forming the aquifers, and argillaceous beds forming aquitards. The shallow aquifers are phreatic to semi-confined, while the deeper aquifers are confined owing to the presence of thick shale/clay beds. The depth to groundwater level was measured in shallow wells (fitted with hand-pumps) during the pre-monsoon (early June 2008) and post-monsoon (early November 2008) periods. It varied from about 12 to 34 m below ground level (bgl) during pre-monsoon to post-monsoon periods and from about 5 to 30 m bgl during the post-monsoon period. The water levels were relatively deeper in the eastern part. The water level fluctuation between pre-monsoon and post-monsoon periods varied from about 5 to 10 m, and a general increase was observed from west to east. Although the direction of groundwater movement varies locally, the general flow direction is south, i.e., in the same direction as the flow of the Pahaj River [38,39]. Two major flow patterns were identified, one in the north to the eastern direction along the flow path of the main river, and the other in the west to east direction.

**Table 1.** Geology of the study area (source: [40]).

| Age | Formation | Lithology |
| --- | --- | --- |
| Recent to Sub-Recent | Soil and alluvium | Sand, clay, gravel, laterite |
| Permian to Triassic | Kamthi | Sandstone and argillaceous beds |
| Upper Permian | Raniganj | Sandstone and carbonaceous shale |
| Upper Permian | Barren Measure | Ferruginous sandstone and clay |
| Lower Permian | Barakar | Feldspathic sandstone, shale, carbonaceous shale with coal seams |

The study area is divided into three zones (A, B, and C) according to the groundwater flow direction in the northern part, western part, and eastern part. Groundwater flows toward the discharge area, through zone C, zone A, and zone B, respectively (Figure 2). The sampling observation stations, water table contour, river flow direction, groundwater flow direction, and flow through area zones are shown in Figure 2. Physiographically, the high F$^-$ wells are concentrated in the eastern part of the study area along the flow path of groundwater movement (Zone C). The maximum elevation is 504 m amsl, which is in

the northeastern part of the area, while the lowest elevation is 240 m amsl, which is in the central part of the area.

## 2.2. Materials and Methods

### 2.2.1. Groundwater Sampling and Analysis

Groundwater samplings were carried out in three phases (Figure 3a,b): (i) groundwater samples from 83 well-distributed hand-pumps (bore wells) in 39 villages across the study area were collected during the pre-monsoon period (8–20 June 2008), (ii) groundwater samples from the same 81 bore wells (the pre-monsoon sample locations, except for two sample locations due to mechanical problems in the hand-pumps) were collected during the post-monsoon period (1–7 November 2008), and (iii) only 20 selective samples around the high-F⁻-incidence area during the mid-monsoon period. The concept of data collection during the mid-monsoon period is not often applied due to extensive effort needed during the rainy season. However, in this study, attempts were made for selective sampling of 20 wells from villages where a high content of fluoride was reported in the pre-monsoon period in order to analyze the dilution effect of high $F^-$ concentration in the mid-monsoon to post-monsoon periods. The depth to water level (DWL) was measured for all the wells during the pre- and post-monsoon periods, and elevation heights at bore wells with respect to mean sea level (MSL) were also measured using a hand-held GPS (GARMIN Etrax, +/−3 m accuracy).

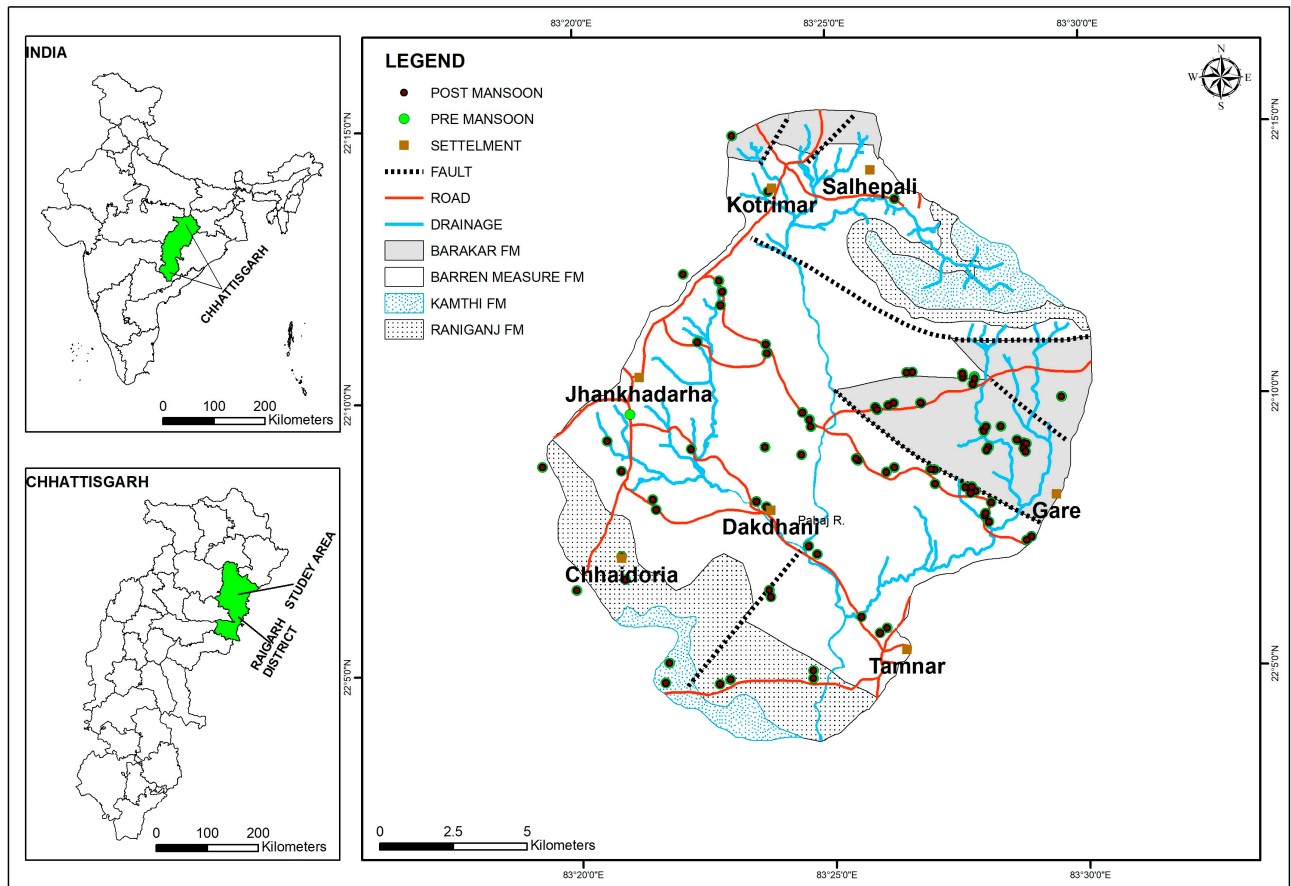

**Figure 1.** Location and geology of the study area with groundwater sampling locations. Source: adopted from [40].

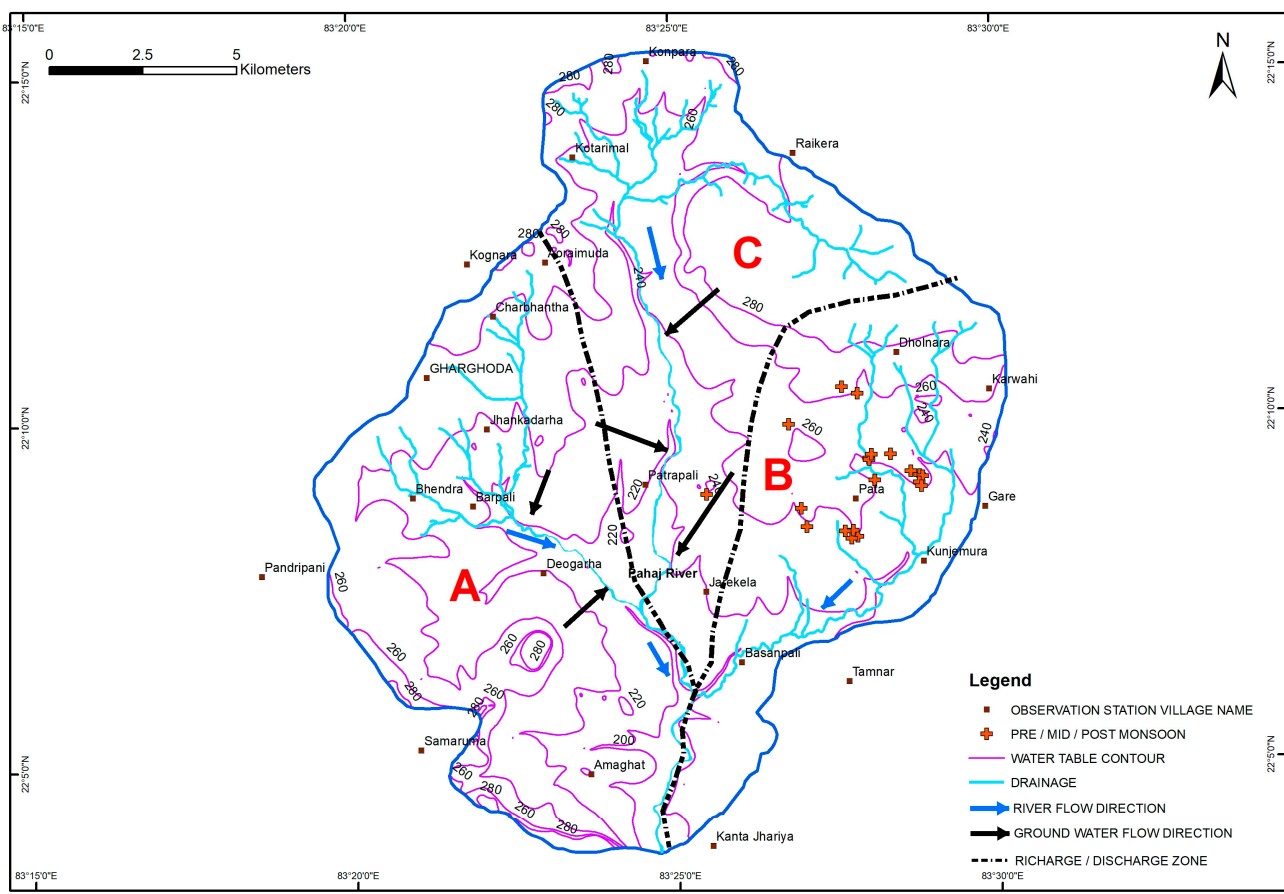

**Figure 2.** Recharge and discharge area (A, B, C) and groundwater flow direction map of the study area.

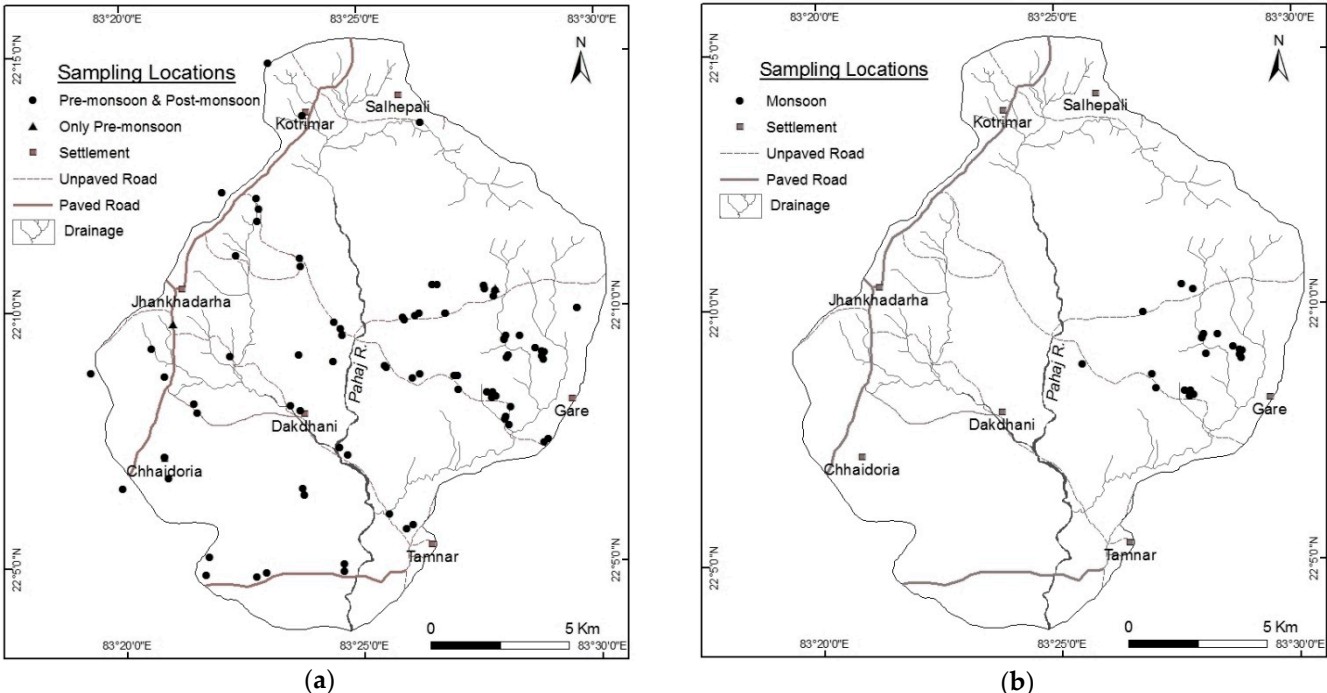

**Figure 3.** Groundwater sampling locations during (**a**) pre-monsoon and post-monsoon periods, and (**b**) the mid-monsoon period.

The groundwater samples were analyzed for 16 hydrochemical parameters, i.e., pH, electrical conductivity (EC), calcium ($Ca^{2+}$), magnesium ($Mg^{2+}$), sodium ($Na^+$), potassium ($K^+$), bicarbonate ($HCO_3^-$), carbonate ($CO_2^-$), sulfate ($SO_4^{2-}$), chloride ($Cl^-$), nitrate ($NO_3^-$), fluoride ($F^-$), phosphate ($PO_4^{3-}$), and total hardness (TH).

Groundwater samples were collected from bore wells after pumping out (hand-pump) for several minutes until constant temperature and conductivity was established and then subsequently filtered through a 0.45 μm membrane filter. Essential physiochemical field parameters such as pH, EC (electrical conductivity), temperature, and TDS (total dissolved solids) were measured by a portable hand-held meter. The water samples for major cation analysis were preserved by adding HNO3 to reduce the pH to ~2. Collected water samples were preserved in polypropylene bottles, stored in a container on ice, and transported to the lab as soon as the in situ parameters were measured. The collected water samples of the pre-monsoon period were analyzed at Ravishankar Shukla University, Dehradun. Total hardness (TH) was measured using a titrimetric method with standard (0.1 M) ETDA solution (ethylenediaminetetraacetic acid). Alkalinity ($CaCO_3$, $HCO_3^-$, and $CO_3$) were calculated by titration with $H_2SO_4$. Sodium ($Na^+$) and potassium ($K^+$) were analyzed using a Systronics flame photometer. Calcium ($Ca^{2+}$) and magnesium ($Mg^{2+}$) ion concentrations were determined through titrimetric analysis using standard ETDA. Sulfate ($SO_4^{2-}$), nitrate ($NO_3^-$), fluoride ($F^-$), and chloride ($Cl^-$) were analyzed using a Systronics spectrophotometer. The cation and anion concentrations for post-monsoon samples were analyzed in an ion chromatograph (Metrohm, 861, Advanced Compact IC) barring the $HCO_3^-$ content, which was determined titrimetrically. The analytical results were found to be acceptable as the ionic charge balance for most of the samples was within the acceptable limit of ±10 [41]. Only two of the 83 pre-monsoon samples and only three of the 81 post-monsoon samples indicated an ionic charge balance beyond the acceptable limit, which were discarded from further analysis. Furthermore, four rock samples (sandstone of Barakar Formation) were also collected from the high-$F^-$-incidence zone during the field campaign. Among the four rock samples, three were surface samples and one was a subsurface sample (from 135 m below ground surface, collected during the drilling of a new bore well). X-ray diffractometry (XRD) and optical microscopy of the sandstones of Barakar Formation were carried out at Wadia Institute of Himalayan Geology, Dehradun, in order to understand the geological controls on $F^-$ distribution in groundwater. The rocks exposed in the eastern part of the study area showed high amounts of muscovite in the hand specimen (Figure 4a), and this was supported by thin section petrographic analysis (Figure 4b). The geological map, groundwater sampling locations, and high-$F^-$-incidence zone were analyzed together through a simple overlay in GIS.

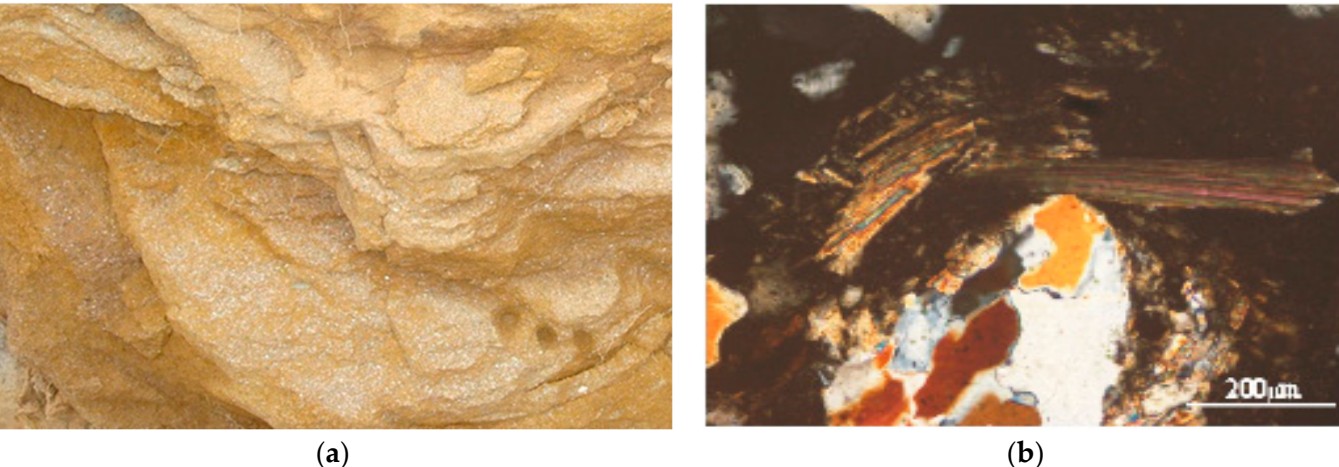

(**a**)      (**b**)

**Figure 4.** (**a**) Field rock sample showing muscovite layer in Barakar Sandstone; (**b**) biotite under thin section (X nicol).

2.2.2. Saturation Index (SI)

The saturation index of groundwater is a measurement that determines if the water is in equilibrium, or if it is undersaturated or oversaturated with certain minerals it may encounter. It helps to identify if the groundwater has the potential to dissolve minerals from the aquifer it traverses, or if it can precipitate minerals back into the water. In simple terms, the saturation index gives an idea of the groundwater's tendency to corrode or form scales on surfaces it interacts with, such as pipes, wells, and equipment. This is crucial in managing water quality, safeguarding infrastructure, and ensuring safe and efficient water usage for industrial, agricultural, or domestic needs [11,42].

The saturation index is often calculated for specific minerals that are relevant to the local geology and hydrogeology. In this instance, we focused on the fluoride concentration in groundwater, which can either dissolve or precipitate depending on certain conditions. To calculate the saturation index for fluoride, the same formula used for other minerals is applied.

$$SI = \log 10(\frac{IAP}{Ksp})$$

SI is the saturation index.
IAP is the ion activity product of fluoride ions in the groundwater.
Ksp is the solubility product constant of fluorite ($CaF_2$).
Fluoride concentration can be affected by the presence of other ions, such as calcium and hydroxide ions, which can interact with fluoride ions and influence its solubility.

SI = 0: the groundwater is in equilibrium with fluorite; as a result, there should be no further dissolution or precipitation of fluorite anticipated.

SI < 0: the groundwater is undersaturated with respect to fluoride, which means that the fluoride ion concentration in groundwater is below the equilibrium concentration, and there is potential for fluorite to dissolve into the water.

SI > 0: that groundwater is oversaturated with respect to fluoride, which means that the fluoride ion concentration in groundwater is above the equilibrium concentration, and there is potential for fluorite to precipitate from the water.

It is crucial to understand that the saturation index only reveals the thermal tendency of minerals to dissolve or precipitate. Other significant factors, such as kinetics, mixing, and complex interactions, may occur in natural groundwater systems, which the index does not account for. Evaluating the saturation index for fluoride in groundwater can aid in the management of drinking water quality and prevent situations where fluoride concentration surpasses recommended limits, which could pose health risks to those who drink the water.

2.2.3. Spatial Distribution Map of Fluoride Concentration in Groundwater

$F^-$ concentration in groundwater has both beneficial and harmful effects on human health. It is harmful if available above or below the permissible level. Low $F^-$ concentration in groundwater causes dental caries, whereas high $F^-$ concentrations can cause dental and skeletal fluorosis.

The lower and upper limits of different classes of $F^-$ concentration were based on box-and-whisker plots. Maps of spatial distribution of fluoride were prepared in the GIS by interpolating the point values of $F^-$ using the inverse distance weighting (IDW) method. The spatial maps showed different fluoride concentration zones: excessive $F^-$ concentration, normal $F^-$ concentration, and low $F^-$ concentration.

2.2.4. Analysis of Geochemical Data

Groundwater quality was determined using the above-mentioned hydrochemical parameters. Basic statistics (mean, median, mode, variance, and standard deviation) of all 16 hydrochemical parameters were computed. The results of the chemical quality of groundwater based on different quality parameters are difficult to interpret. To overcome this, a graphical representation is quite useful. One of the widely used graphs for displaying,

representing, and comparing water quality analysis is the trilinear diagram by Piper [43]. In the Piper plot, both cations and anions are plotted as a percentage of milli-equivalents in two base triangles. The total cations and anions in meq/L are set equal to 100%. The data points in the two base triangles are then projected onto the central diamond shaped grid parallel to the upper edges of the central area. The projection indicates the similarities and differences among the samples. Those with similar properties tend to plot together as a group.

2.2.5. Relationship between Fluoride Concentration and Different Hydrochemical Parameters

Fluoride is highly reactive chemical element, and its concentration in groundwater depends on several factors such as pH, solubility of $F^-$-bearing minerals, anion-exchange capacity of aquifer materials, and geological, physical, and chemical composition of aquifers [44,45].

Fluoride concentration in groundwater has been found to be correlated with different hydrochemical parameters. A positive correlation between $F^-$ and silica, and between $F^-$ and sodium in groundwater indicates a silicate mineral source of $F^-$ [31] whereas a negative correlation was reported between $F^-$ and calcium [46]. Low calcium and high bicarbonate in groundwater favor fluoride concentration in groundwater [7]. Hence, in order to understand the dynamics and relationship of fluoride concentration in groundwater, the above-mentioned hydrochemical parameters were analyzed for their correlation with fluoride concentration using univariate and multivariate statistical methods and a geographical information system (GIS) [47].

## 3. Results

### 3.1. Analysis of Hydrochemical Properties of Groundwater

A summary of the hydrochemical analysis of fluoride ($F^-$), pH, electrical conductivity (EC), calcium ($Ca^{2+}$), magnesium ($Mg^{2+}$), sodium ($Na^+$), potassium ($K^+$), bicarbonate ($HCO_3^-$), carbonate ($CO^{2-}$), sulfate ($SO_4^{2-}$), chloride ($Cl^-$), nitrate ($NO_3^-$), phosphate ($PO_4^{3-}$), and total hardness (TH) in the pre-monsoon and post-monsoon periods are shown in Table 2. Details of the hydrochemical properties of the groundwater and their relationship with fluoride concentration are discussed in Section 3.4.

**Table 2.** Summary of hydrochemical characteristics of groundwater. Units are in mg/L except for pH and EC (electrical conductivity, µS/cm).

| Parameter | Pre-Monsoon Period (Number of Samples = 83) | | | | Post-Monsoon Period (Number of Samples = 81) | | | |
|---|---|---|---|---|---|---|---|---|
| | Range | Mean | Median | SD | Range | Mean | Median | SD |
| pH | 6.91–8.96 | 8.18 | 7.02 | 0.49 | 6.36–7.85 | 7.02 | 7.02 | 0.49 |
| EC | 78–2760 | 441.1 | 380.0 | 333.9 | 95–1268 | 453.3 | 400.0 | 226.1 |
| $Ca^{2+}$ | 9.9–130.7 | 32.3 | 25.7 | 20.6 | 4.8–62.7 | 26.2 | 22.3 | 13.9 |
| $Mg^{2+}$ | 2.4–125.2 | 20.0 | 16.9 | 16.0 | 2.0–42.2 | 13.1 | 11.0 | 8.3 |
| $Na^+$ | 0.6–62.7 | 18.2 | 16.9 s | 12.9 | 1.0–84.7 | 16.4 | 14.0 | 13.6 |
| $K^+$ | 3.0–80.5 | 21.3 | 17.4 | 15.7 | 1.1–44.9 | 14.9 | 11.7 | 10.6 |
| $Li^+$ | nm | - | - | - | 0–0.08 | 0.01 | 0.0 | 0.02 |
| $HCO_3^-$ | 24.4–483.6 | 190.9 | 186.5 | 88.0 | 30.3–519.4 | 198.3 | 182.3 | 93.9 |
| $CO_3^{2-}$ | nd | - | - | - | 0–29.2 | 2.4 | 0.0 | 5.8 |
| $SO_4^{2-}$ | 1.5–31.8 | 10.6 | 9.6 | 5.7 | 0–215.0 | 6.8 | 2.9 | 23.9 |
| $Cl^-$ | 0–335.8 | 24.0 | 12.0 | 45.9 | 1.0–90.4 | 15.4 | 10.9 | 17.0 |
| $NO_3^-$ | 0–106.3 | 4.2 | 0.5 | 14.0 | 0–36.4 | 2.2 | 0.5 | 5.4 |
| $F^-$ | 0.09–8.88 | 1.08 | 0.64 | 1.6 | 0–7.12 | 1.03 | 0.5 | 1.52 |
| $PO_4^{3-}$ | nd | - | - | - | nd | - | - | - |
| $SiO_2$ | 10–150 | 58.7 | 60.0 | 31.5 | nm | - | - | - |
| TH | 34.7–841 | 164.9 | 143.5 | 113.3 | 44.6–401 | 163.2 | 146.0 | 81.1 |

TH—total hardness; nd—not detectable; nm—not measured.

### 3.2. Analysis of Fluoride Concentration in Groundwater

Both high and low concentrations of fluoride concentration in groundwater have a detrimental impact on human health. The WHO [36] recommends 1.0 mg/L as the desirable limit and 1.5 mg/L as the maximum allowable limit of $F^-$ in drinking water sources. However, in the Indian context, the limits of $F^-$ concentrations are fixed with respect to desirable and maximum allowable limits in drinking water sources between 0.6 and 1.2 mg/L [48].

The $F^-$ analysis results in the study area revealed that the concentration of $F^-$ in groundwater in pre-monsoon periods varied from 0.09 to 8.88 mg/L (mean: 1.08 mg/L, median: 0.64 mg/L); in post-monsoon period, it fluctuated between 0.01 and 7.12 mg/L (mean: 1.03 mg/L, median: 0.5 mg/L).

Among the total samples, around 15% (12 samples) and 16% (13 samples) of the pre- and post-monsoon samples, respectively, had fluoride concentrations above the desirable limit ($F^- > 1.2$). In addition, in 10 pre-monsoon (out of 12 samples) and 11 post-monsoon (out of 13 samples) samples, the $F^-$ concentration exceeded the maximum permissible limit of 1.5 mg/L. At six locations, the $F^-$ concentration was even higher than 3 mg/L and reached up to 8.8 mg/L for pre-monsoon and 7.1 mg/L for post-monsoon conditions. Selective sampling (N = 20) and analysis in and around the high-$F^-$-incidence zone accomplished during the monsoon period confirmed $F^-$ concentrations higher than the desirable limit (1.2 mg/L) for the same wells (Table 3).

**Table 3.** The number of samples in different $F^-$ concentration ranges (mg/L) during the pre-monsoon and post-monsoon periods.

| Sampling Time | Number of Samples and Concentration Value in mg/L | | | |
|---|---|---|---|---|
| | $F^- < 0.6$ | $0.6 \leq F^- \leq 1.2$ | $F^- > 1.2$ | $F^- > 1.5$ |
| Pre-monsoon period (N = 83) | 40 (48%) | 31(37%) | 12 (15%) | 10 (12%) |
| Post-monsoon period (N = 81) | 54 (67%) | 14 (17%) | 13 (16%) | 11 (13%) |
| Mid-monsoon period * (N = 20) | 5 | 2 | 13 | 10 |

* Selective sampling during the mid-monsoon period was only performed in and around the high-$F^-$—incidence zone.

Around 48% (40 samples) of pre-monsoon samples and 67% (54 samples) of post-monsoon samples showed fluoride concentrations below the minimum required level. The remaining 37% (31 samples) of pre-monsoon and 17% (14 samples) of post-monsoon samples had $F^-$ concentrations within the optimum range (i.e., 0.6–1.0 mg/L).

### 3.3. Saturation Index of Fluoride in Groundwater

Geochemical modeling was conducted to determine the chemical equilibrium within the sedimentary aquifer. The saturation indices for anhydrite, aragonite, calcite, dolomite, fluorite, gypsum, and halite were calculated, and the results were plotted for pre-monsoon samples (Figure 5) and post-monsoon samples (Figure 6). Calcite and fluorite are important minerals in the context of fluoride mobilization [49]. All samples were found to be undersaturated in the pre-monsoon period with respect to calcite, fluorite, halite, gypsum, anhydrite, and dolomite. This suggests that the minerals can dissolve more in groundwater; hence, the possibility of their increasing concentration in groundwater is greater. Halite is in a dissolution state; it increases $Na^+$ in groundwater, which favors an increase in $F^-$ content in groundwater. An increase in $Na^+$ ion concentration can promote the cation-exchange process with $Ca^{2+}$, which in turn increases the $F^-$ concentration in groundwater. On the other hand, in the post monsoon period, calcite and dolomite were found to be oversaturated, preventing further dissolution; therefore, it would precipitate as $CaF_2$. On the other hand, fluorite was in an undersaturated state due to calcite oversaturation, reducing calcium

activity and allowing more fluorite to dissolve, which increased the $F^-/Ca^{2+}$ concentration of the solution, in line with the study of Alamry [50].

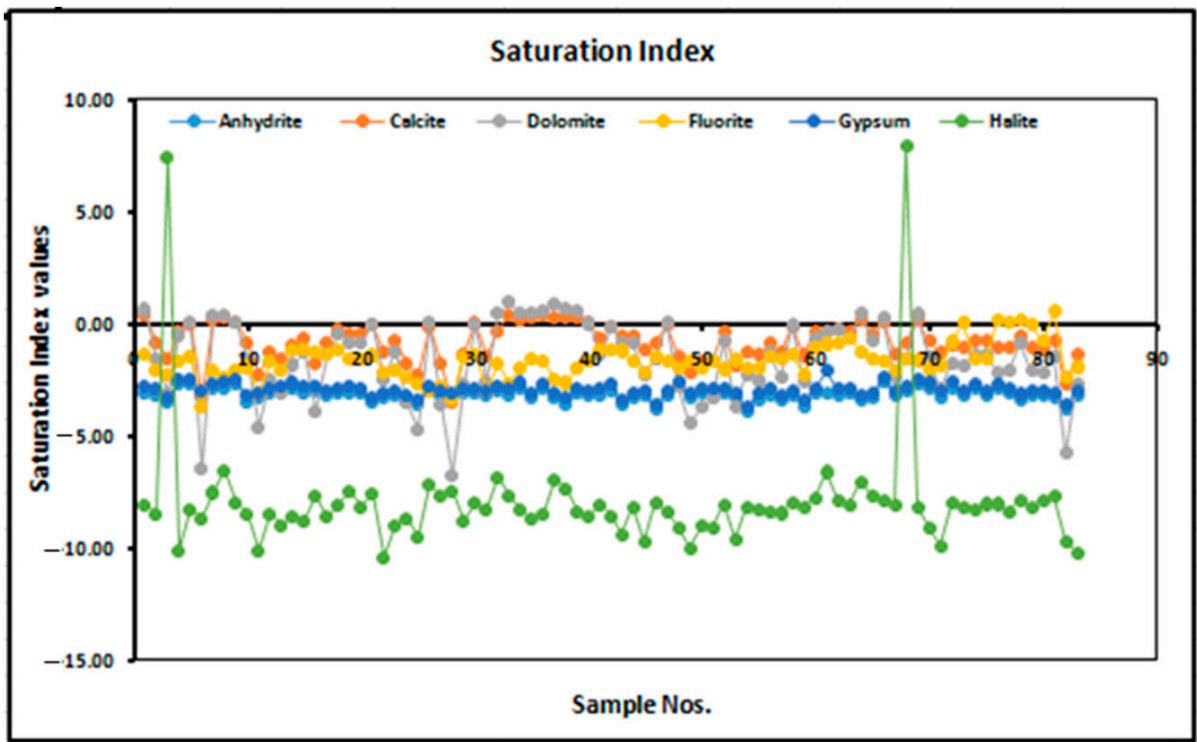

**Figure 5.** SI values in pre-monsoon groundwater samples (N = 81).

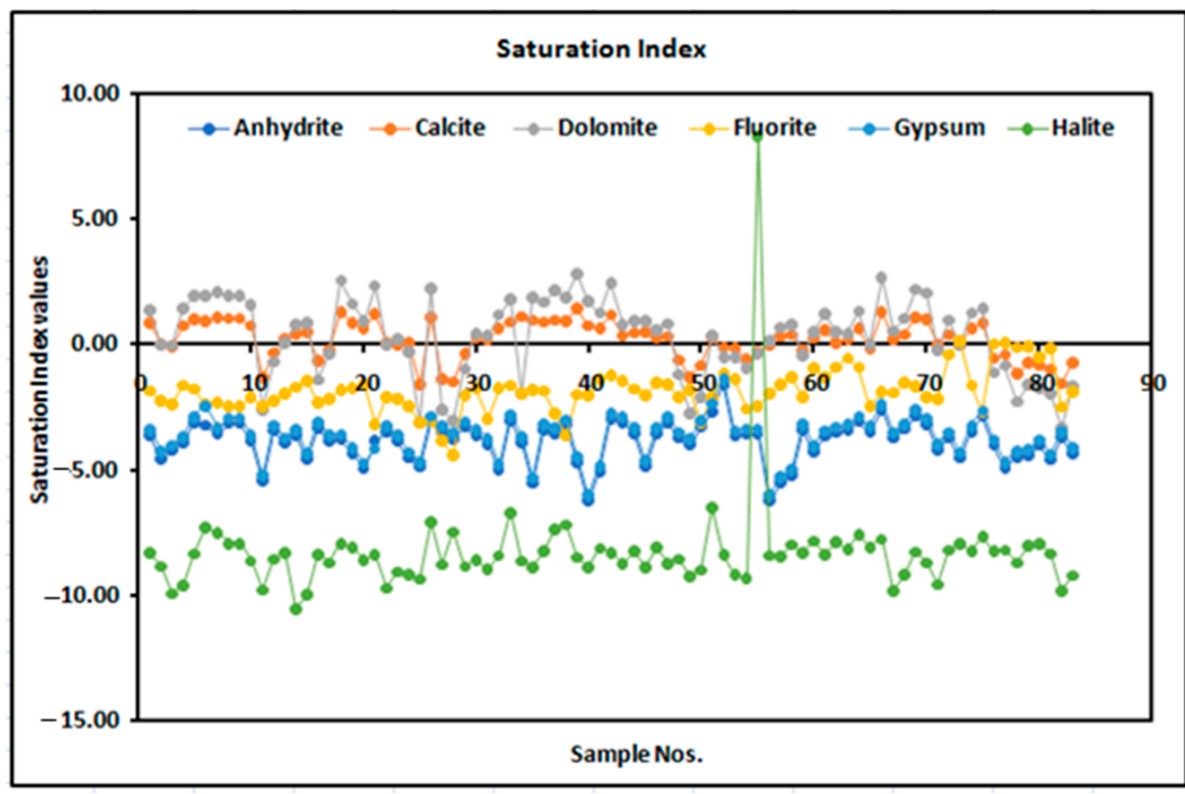

**Figure 6.** SI values in post-monsoon groundwater samples (N = 81).

### 3.4. Spatio-Temporal Distribution of F⁻ in Groundwater

Spatial distribution maps of fluoride were prepared by interpolating the point values (83 samples collected during pre-monsoon period and 81 samples collected during post-monsoon period) using the inverse distance weighting (IDW) method in a GIS environment.

Areas with $F^-$ content less than 0.6 mg/L can lead to dental caries and poor bone development; those areas with content ranging between 0.6 and 1.0 mg/L are classified under the safe category; those with $F^-$ concentration ranging between 1.0 and 3.0 mg/L pose a risk of dental fluorosis; those with $F^-$ concentrations higher than 3.0 mg/L are highly contributory to dental and skeletal fluorosis.

The spatial maps of fluoride concentration in the pre-monsoon and post-monsoon periods (Figure 7a and 7b, respectively) indicate a very high concentration of fluoride (3.0–8.8 mg/L and 3.0–7.1 mg/L) and a high concentration of 1.2–3.0 mg/L (above the permissible limit) in the eastern part of the study area. This shows a serious health risk for the population residing in the fluoride hotspot areas. The field campaigns also confirmed that the population residing in the area where the wells showed high $F^-$ concentrations (above the desirable and permissible limits) were prone to dental and skeletal fluorosis. Five villages in this zone are Muragaon, Saraitola, Pata, Kunjhemura, and Dolnara. It is further revealed from Figure 7 that, in addition to these villages, populations living in adjacent villages, e.g., Gare, Rodopalli, Regaon, Manjhapara, Bajarmura, and Karuwahi, are also potentially at risk.

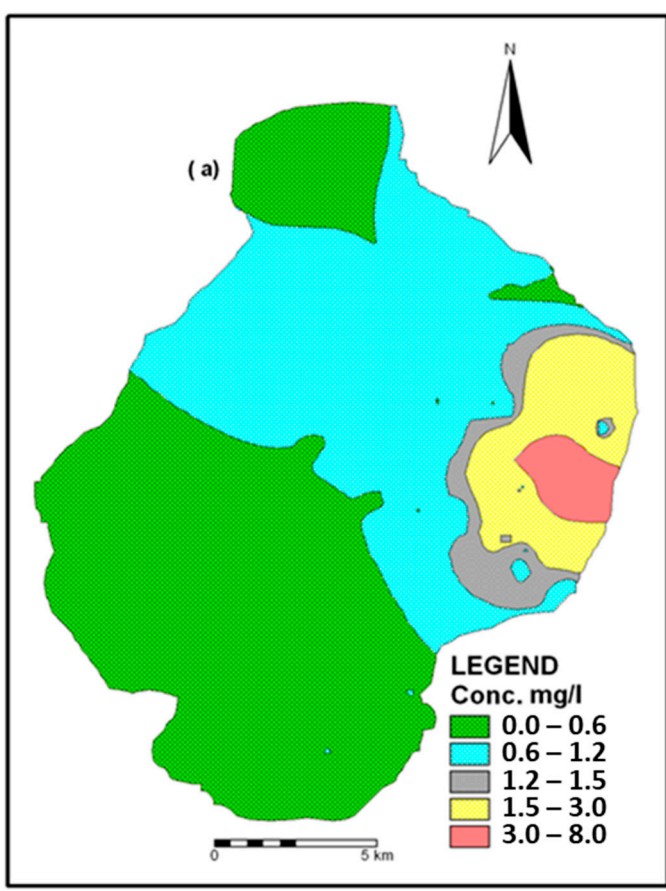 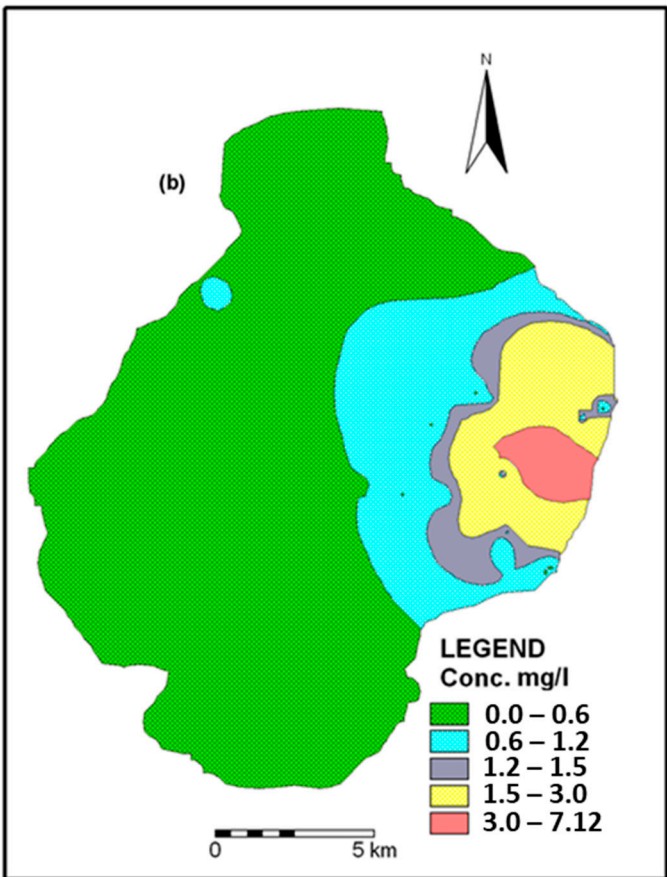

**Figure 7.** Spatial distribution of F⁻ in groundwater during pre-monsoon (**a**) and post-monsoon (**b**) periods.

It was also found that a major proportion of the study area in the southwestern part (during the pre-monsoon period) and a comparatively greater proportion of the study area representing the southern, western, and northern parts during the post-monsoon period showed fluoride concentrations between 0.0 and 0.6 mg/L (below the permissible limit). Low values of fluoride concentration in the groundwater also have a harmful effect on human health, and these areas should be supplemented with extra fluoride intake.

To understand the temporal variation of $F^-$ levels in groundwater, in the study area, a number of samples in different $F^-$ concentration ranges during the pre-monsoon, mid-monsoon, and post-monsoon periods were analyzed (Table 3). The comparison of pre-monsoon and post-monsoon data (Tables 2 and 3) suggests a dilution effect owing to fresh recharge on account of monsoon rainfall. However, it is important to note that the dilution effect does not cause any appreciable change in the high-$F^-$-incidence zone (Figure 7b).

### 3.5. Groundwater Types vis-à-vis Fluoride in Groundwater

The groundwater samples were more or less alkaline, with pH values varying from 6.91 to 8.96 with a mean of 8.16 in the pre-monsoon period, and from 6.36 to 7.85 with a mean of 7.02 in the post-monsoon period (Table 2). The presence of high pH favors the release of $F^-$ from the aquifer matrix into groundwater [51,52].

The ionic concentration of major cations and anions in pre- and post-monsoon samples are plotted in Piper's trilinear diagram [53] (Figure 8a,b). In the pre-monsoon period, the samples belonged to three major groundwater types—(i) $Ca-Mg-HCO_3$ type, (ii) $Ca-Mg-Cl$ type, and (iii) mixed type (i.e., no dominant type of water). In the post-monsoon period, in addition to the above three groundwater types, the $Ca-Mg-SO_4$ type occurred. The majority of groundwater samples belonged to the $Ca-Mg-HCO_3$ type, wherein fluoride concentration ranged between the desired and maximum permissible limit ($0.6 \leq F^- \leq 1.2$ mg/L). Fluoride concentration was high (i.e., >1.2 mg/L) mainly in the mixed water types ($Na-Ca-HCO_3$, $Na-Ca-Mg-HCO_{3'}$ and $Na-Mg-Ca-HCO_3$), whereas $Na^+$ concentration was relatively higher than other cations compared to water of the $Ca-Mg-HCO_3$ type (Figure 8). Moreover, 27% of groundwater samples of pre-monsoon period belonged to the mixed type, likely associated with the cause of fluoride dissolution in the study area. Geochemical studies previously performed suggested that a $NaHCO_3$ water type, alkaline nature of water, low calcium concentration, and high sodium concentration are favorable conditions for accelerating the dissolution process that is responsible for high $F^-$ in groundwater [16,28–30,54].

In order to understand the relationship between $F^-$ concentration and groundwater type, three sets of Piper diagram were plotted by dividing the water samples into three classes according to Indian drinking water standard: (1) $F^- < 0.6$ mg/L (2) $F^-$ ranging between 0.6–1.2 mg/L, and (3) $F^- > 1.2$ mg/L (Figure 8c).

In order to compare the $F^-$ content and groundwater types in the post-monsoon water samples, three sets of piper diagram were plotted by dividing the water samples into 3 classes as shown in Figure 8d. The classification of $F^-$ content in groundwater proposed by ISI (1983) was used to compare the $F^-$ content in different water groups. The $F^-$ content in each type of water varied. The lowest $F^-$ content ($0.6 < F^- < 1.2$ mg/L) occurred in the $Ca-Mg-HCO_3$, $Ca-Mg-HCO_3-Cl$, and $Ca-Mg-SO_4$ types of water. A high $F^-$ concentration was associated with mixed water types, as described below (Figure 8d).

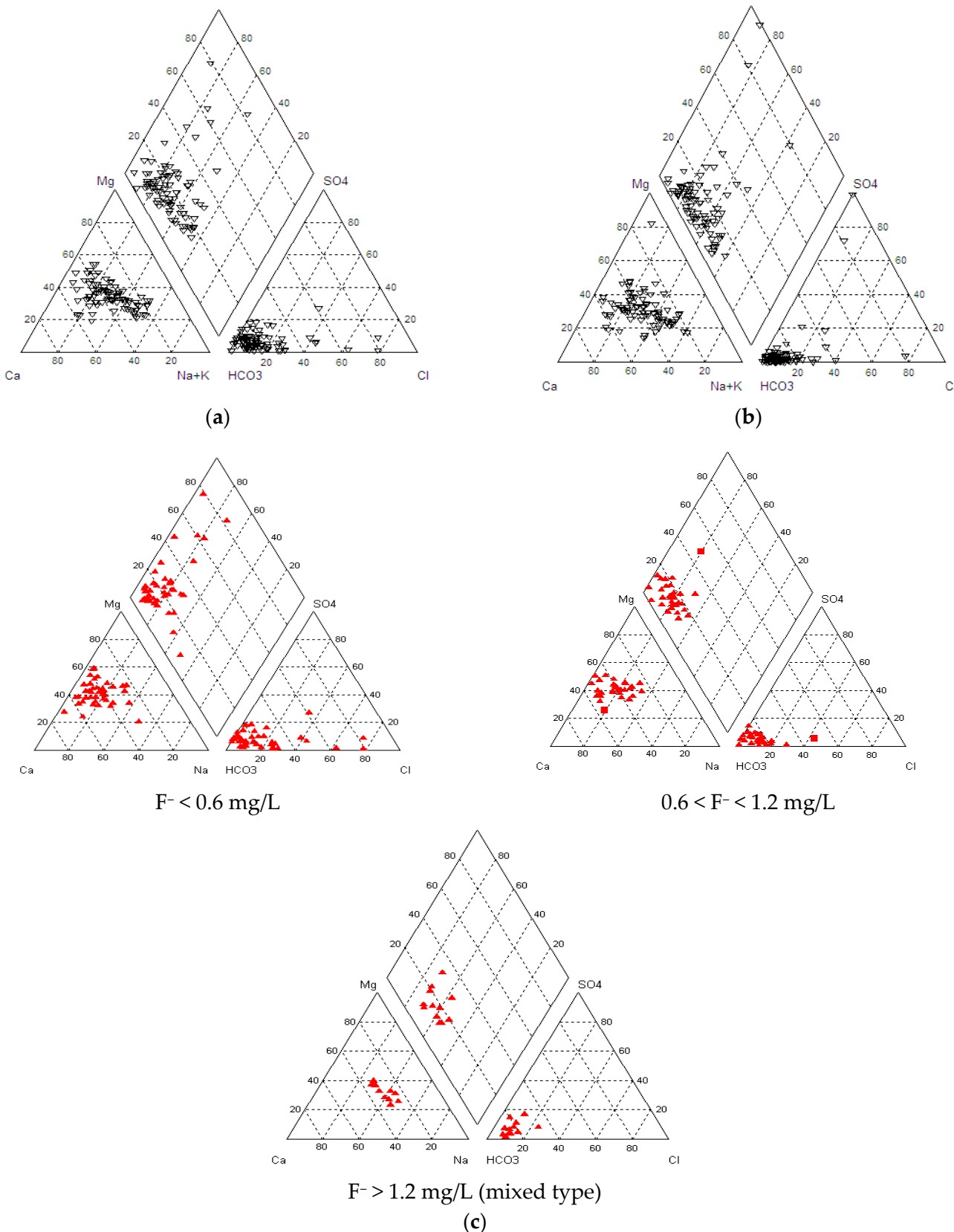

Figure 8. *Cont.*

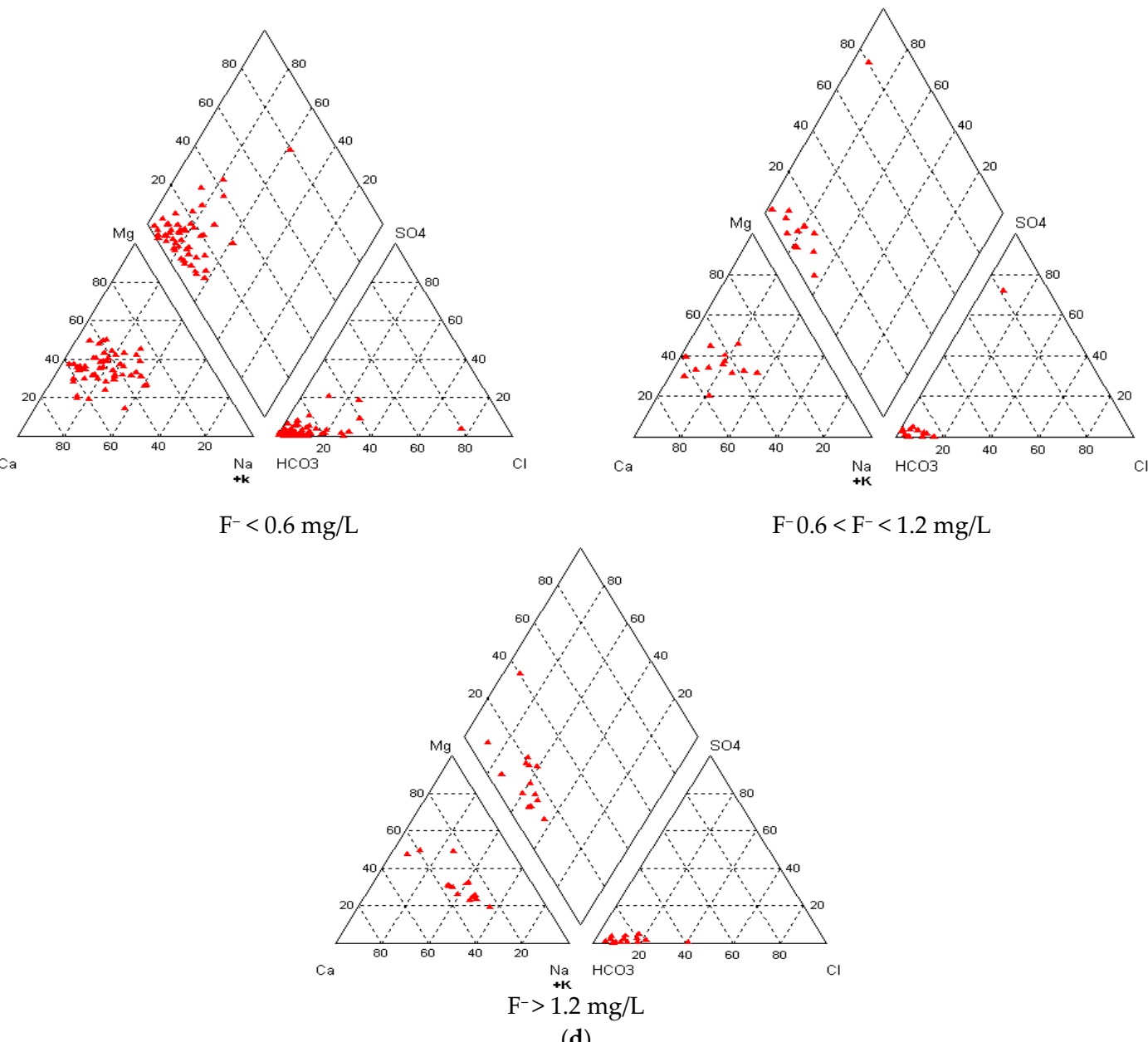

**Figure 8.** Piper's trilinear diagrams showing major ion chemistry of groundwater samples: (**a**) premonsoon period; (**b**) post-monsoon period. (**c**) Piper diagrams showing water types with respect to F⁻ content (pre-monsoon period). (**d**) Piper diagram showing F⁻ water types with respect to F⁻ content (post-monsoon period).

Apambire et al. [44] reported that groundwater with high F⁻ concentration is generally of Na-HCO$_3$ type with low Ca$^{2+}$ concentration. To explore the validity of this relationship in the present study, pie diagrams were plotted for major cations (Figure 9) and for major anions (Figure 10) according to two classes of water samples: (i) F⁻ ≤ 1.2 mg/L, and (ii) F⁻ > 1.2 mg/L. In groundwater samples with F⁻ ≤ 1.2 mg/L, Ca$^{2+}$ and Mg$^{2+}$ were the dominant cations, followed by Na$^+$ and K$^+$. In groundwater samples with F⁻ > 1.2 mg/L, Na$^+$ was the dominant cation (its concentration nearly doubled), followed by Ca$^{2+}$ and Mg$^{2+}$ (Figure 9). The concentration of K$^+$ remained nearly the same in both pre- and

post-monsoon periods. The ratio between $Na^+$ and $Ca^{2+}$ (i.e., Na:Ca) was nearly three times for groundwater samples having $F^- > 1.2$ mg/L as compared to that for groundwater samples having $F^- \leq 1.2$ mg/L (i.e., 0.42 vs. 1.16 in the pre-monsoon period and 0.46 vs. 1.32 in the post-monsoon period). As discussed earlier, an increase in $F^-$ concentration is generally associated with an increase in $HCO_3^-$ concentration. This relationship was not observed, however, in the study area because there was no appreciable change in $HCO_3^-$ concentration with an increase in $F^-$ concentration.

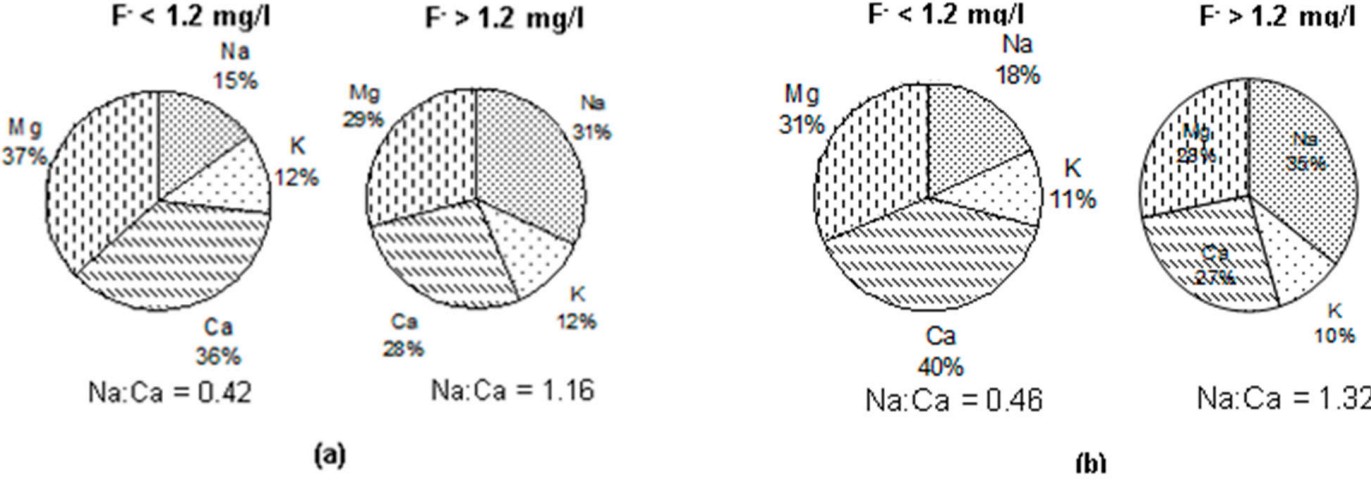

**Figure 9.** Pie diagrams showing relationship between $F^-$ and major cations: (**a**) pre-monsoon period; (**b**) post-monsoon period. The relative proportions of ions are based on their concentrations in mg/L.

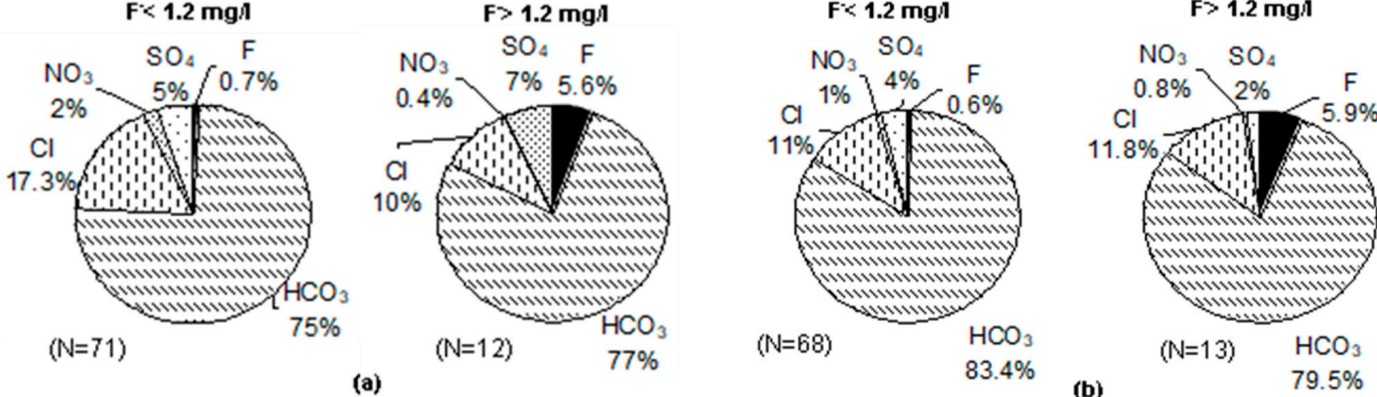

**Figure 10.** Pie diagrams showing relationship between $F^-$ and major anions: (**a**) pre-monsoon period; (**b**) post-monsoon period. The relative proportions of ions are based on their concentrations in mg/L.

### 3.6. Relationship between Fluoride and Other Hydrochemical Parameters

Linear plots offer a visualization of the statistical relationship of fluoride with other geochemical parameters and aid to deduce the causative and controlling factors, as well as processes responsible for the enrichment of $F^-$ in groundwater [30,55]. In order to review the relationship of $F^-$ with other hydrochemical parameters, correlation matrices were prepared for the pre-monsoon (Table 4) and post-monsoon (Table 5) periods.

**Table 4.** Correlation matrix of different hydrochemical parameters in groundwater during the pre-monsoon period.

| | pH | EC | TH | $Ca^{2+}$ | $Mg^{2+}$ | $Na^+$ | $K^+$ | $HCO_3^-$ | $SO_4^{2-}$ | $Cl^-$ | $NO_3^-$ | $F^-$ | $SiO_2$ |
|---|---|---|---|---|---|---|---|---|---|---|---|---|---|
| pH | 1 | | | | | | | | | | | | |
| EC | 0.28 ** | 1 | | | | | | | | | | | |
| TH | 0.37 ** | 0.95 ** | 1 | | | | | | | | | | |
| $Ca^{2+}$ | 0.44 ** | 0.89 ** | 0.95 ** | 1 | | | | | | | | | |
| $Mg^{2+}$ | 0.31 ** | 0.96 ** | 0.97 ** | 0.87 ** | 1 | | | | | | | | |
| $Na^+$ | 0.33 ** | 0.43 ** | 0.38 ** | 0.39 ** | 0.36 ** | 1 | | | | | | | |
| $K^+$ | 0.10 | 0.34 ** | 0.22 * | 0.19 * | 0.24 * | 0.16 | 1 | | | | | | |
| $HCO_3^-$ | 0.67 ** | 0.46 ** | 0.58 ** | 0.62 ** | 0.55 ** | 0.56 ** | 0.13 | 1 | | | | | |
| $SO_4^{2-}$ | −0.03 | −0.15 | −0.16 | −0.16 | −0.13 | 0.12 | −0.18 | −0.03 | 1 | | | | |
| $Cl^-$ | 0.10 | 0.92 ** | 0.84 ** | 0.80 ** | 0.82 ** | 0.36 ** | 0.34 ** | 0.17 | −0.20 * | 1 | | | |
| $NO^{3-}$ | −0.54 * | 0.02 | −0.04 | −0.08 | −0.06 | 0.01 | 0.26 ** | −0.29 ** | 0.0 | 0.13 | 1 | | |
| $F^-$ | 0.03 | −0.11 | −0.22 * | −0.22 * | −0.20 * | 0.31 ** | −0.17 | −0.10 | 0.12 | −0.12 | −0.11 | 1 | |
| $SiO_2$ | −0.09 | −0.22 * | −0.30 ** | −0.37 ** | −0.24 * | 0.20 * | −0.14 | −0.18 * | 0.19 * | −0.20 * | −0.18 | 0.36 ** | 1 |

* Statistically significant at 0.05 level; ** statistically significant at 0.01 level.

**Table 5.** Correlation matrix of different hydrochemical parameters in groundwater during the post-monsoon period.

| | pH | EC | TH | $Ca^{2+}$ | $Mg^{2+}$ | $Na^+$ | $K^+$ | $Li^+$ | $HCO_3^-$ | $SO_4^{2-}$ | $Cl^-$ | $NO_3^-$ | $F^-$ |
|---|---|---|---|---|---|---|---|---|---|---|---|---|---|
| pH | 1 | | | | | | | | | | | | |
| EC | 0.25 * | 1 | | | | | | | | | | | |
| TH | 0.31 ** | 0.92 ** | 1 | | | | | | | | | | |
| $Ca^{2+}$ | 0.36 ** | 0.82 ** | 0.88 ** | 1 | | | | | | | | | |
| $Mg^{2+}$ | 0.21 * | 0.85 ** | 0.89 ** | 0.76 ** | 1 | | | | | | | | |
| $Na^+$ | 0.10 | 0.73 ** | 0.52 ** | 0.42 ** | 0.57 ** | 1 | | | | | | | |
| $K^+$ | 0.18 | 0.22 * | 0.08 | 0.06 | 0.04 | 0.04 | 1 | | | | | | |
| $Li^+$ | −0.24 * | 0.08 | 0.05 | −0.07 | 0.14 | 0.12 | 0.12 | 1 | | | | | |
| $HCO_3^-$ | 0.38 ** | 0.81 ** | 0.82 ** | 0.77 ** | 0.86 ** | 0.60 ** | 0.09 | 0.08 | 1 | | | | |
| $SO_4^{2-}$ | −0.04 | 0.20 * | 0.25 * | 0.33 ** | 0.11 | −0.1 | 0.06 | −0.02 | −0.10 | 1 | | | |
| $Cl^-$ | 0.07 | 0.69 ** | 0.51 ** | 0.50 ** | 0.44 ** | 0.66 ** | 0.29 ** | −0.10 | 0.35 ** | 0.06 | 1 | | |
| $NO_3^-$ | 0.03 | 0.18 | 0.07 | 0.14 | 0.06 | 0.22 * | 0.11 | −0.10 | 0.01 | 0.03 | 0.27 ** | 1 | |
| $F^-$ | −0.22 * | −0.22 * | −0.30 ** | −0.33 ** | −0.24 * | 0.21 * | −0.18 * | 0.14 | −0.22 * | −0.05 | −0.10 | −0.04 | 1 |

* Statistically significant at 0.05 level; ** statistically significant at 0.01 level.

### 3.6.1. Pre-Monsoon Period: Correlation between Fluoride Concentrations and Other Hydrochemical Parameters

The correlation matrices exhibited significant positive correlations among EC, TH, $Ca^{2+}$, $Mg^{2+}$, $Na^+$, $K^+$, $HCO_3^-$, and $Cl^-$ whereas positive but poor correlations existed with $SO_4^-$ in the pre-monsoon period (Table 4). A low but significant negative correlation was observed between EC and $SiO_2$ (analyzed only during the pre-monsoon period), which may be explained by the fact that silicate rocks, being more resistant to weathering, contribute less dissolved load. The negative correlation of $F^-$ with $HCO^{3-}$ (significant in the post-monsoon period and insignificant in the pre-monsoon period) was contrary to the general observations. However, such a negative correlation has also been reported by a few researchers from relatively deeper aquifers of basaltic terrain in central India (e.g., [56,57]). The scatter plots of $F^-$ vis-à-vis $Ca^{2+}$, $Na^+$, $HCO^{3-}$, and pH are shown in Figure 11a–d, respectively.

### 3.6.2. Post-Monsoon Period: Correlation between Fluoride Concentrations and Other Hydrochemical Parameters

It can be observed from Tables 4 and 5 that $F^-$ in groundwater had (a) a low but significant positive correlation with $Na^+$ and $SiO_2$, (b) a low but significant negative correlation with $Ca^{2+}$, $Mg^{2+}$, TH, $HCO_3^-$, $K^+$, EC, and pH (for $HCO_3^-$, $K^+$, EC, and pH, the correlation was significant only during the post-monsoon period), (c) a positive but poor correlation with $Li^+$ (analyzed only during the post-monsoon period), (d) a negative but poor correlation with $Cl^-$ and $NO_3^-$, and (e) a poor but negative correlation with $SO_4^{2-}$ in the post-monsoon period.

Plots of fluoride versus pH did not show a significant correlation (Figure 11d), but hydrochemical characteristics indicating the alkaline quality of water (Table 2) accelerate the enrichment of concentration of fluoride in groundwater [58]. A significant positive correlation of wells having a fluoride concentration >1.2 mg/L was noticed between fluorides and sodium (Figure 11b), showing that an alkaline environment is an important regulating process for $F^-$ leaching from fluoride-bearing minerals [59,60]. In addition, as illustrated in Figure 9, the correlation plots of fluoride and calcium strongly indicated that the presence of high concentration of calcium favored low $F^-$ in groundwater. However, $F^-$ had no substantial correlations with $SO_4^{2-}$, $Cl^-$, and $K^+$, probably showing that the proportion of ionic concentration contributed to aquifer material was not from identical sources. No considerable relationship between fluoride and $NO_3^-$ existed, suggesting a geogenic source of fluoride.

### 3.7. Geologic Control on Fluoride Distribution in Groundwater

It was observed that the high-$F^-$-incidence zone in the eastern sector of the study area fell mainly in the Barakar Formation and partly in the Barren Measures Formation just south of its contact with the Barakar Formation. In terms of the number of samples, eight and five samples fell in the Barakar and Barren Measures Formations, respectively (Figure 12a).

The spatial distribution of $F^-$ concentration in groundwater pertaining to lithostratigraphy in the pre- and post-monsoon periods is shown in the form of box-and-whisker plots (Figure 12b). The highest concentration of $F^-$ in groundwater (maximum, mean, and median values) occurred in the Barakar Formation, followed by the Barren Measures Formation, Kamthi Formation, and Raniganj Formation. It was, however, noted from detailed analysis of geological sections and well depths that the wells with high $F^-$ located superficially on the Barren Measures Formation actually tapped into the underlying aquifers of the Barakar Formation. Thus, the lithological/mineralogical assemblage of the coal-bearing Barakar Formation appeared to be a dominant source of high $F^-$ concentration in the bore wells located in the study area (Figure 12a).

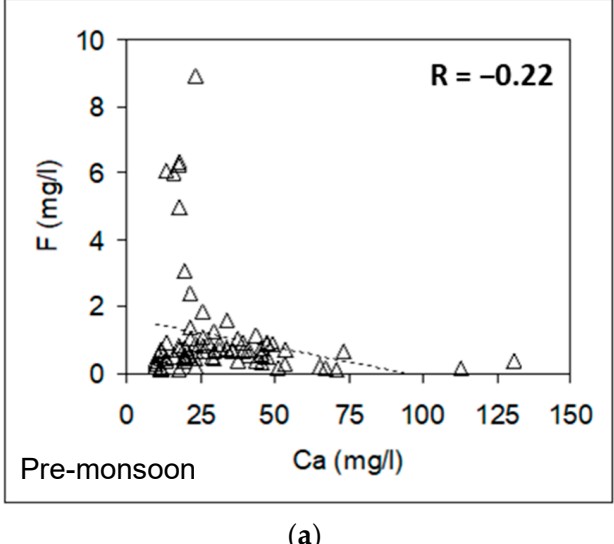
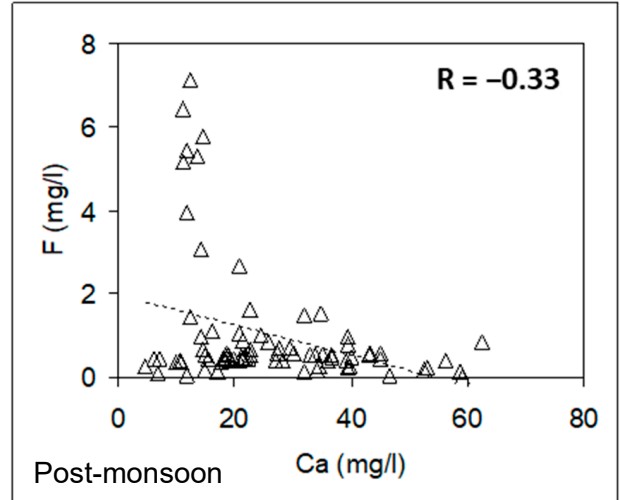

(**a**)

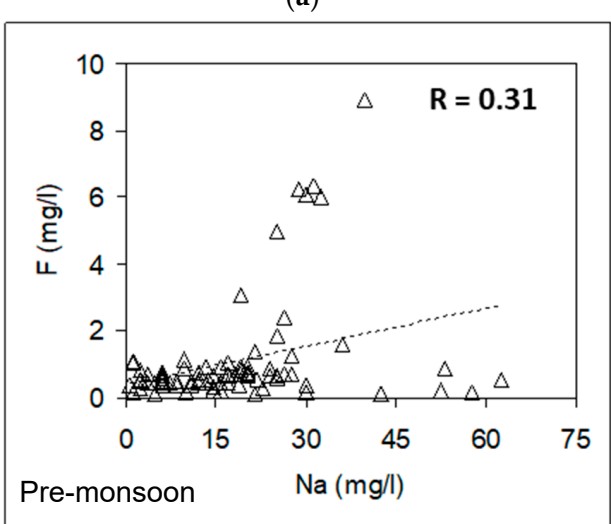
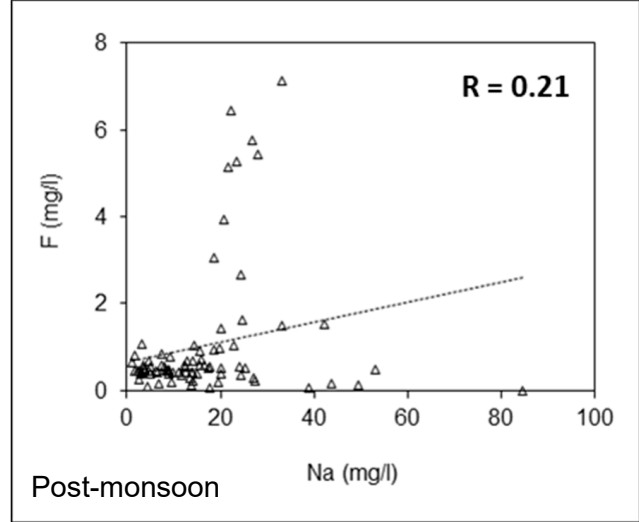

(**b**)

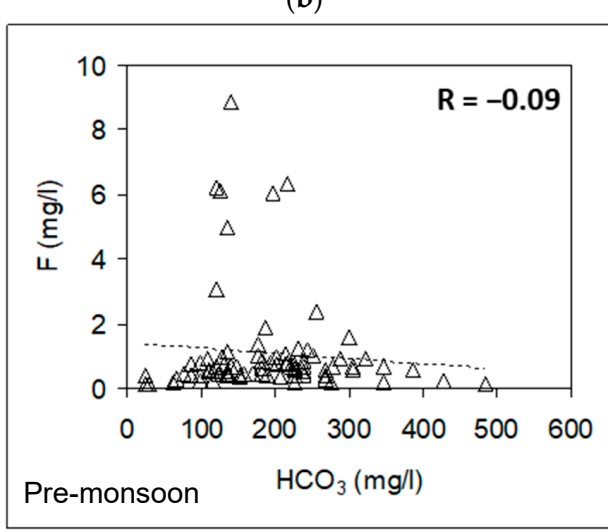
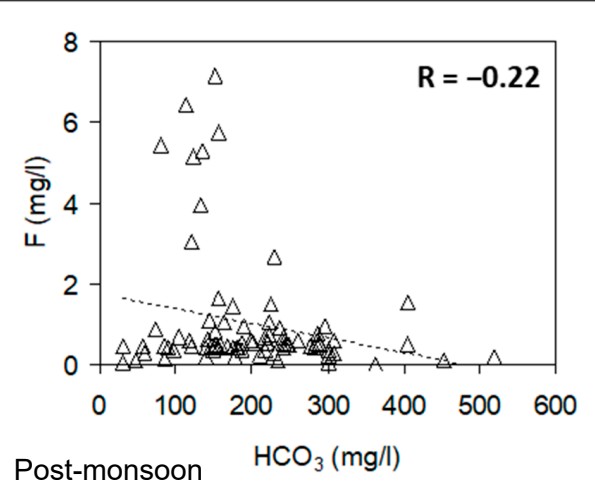

(**c**)

**Figure 11.** *Cont.*

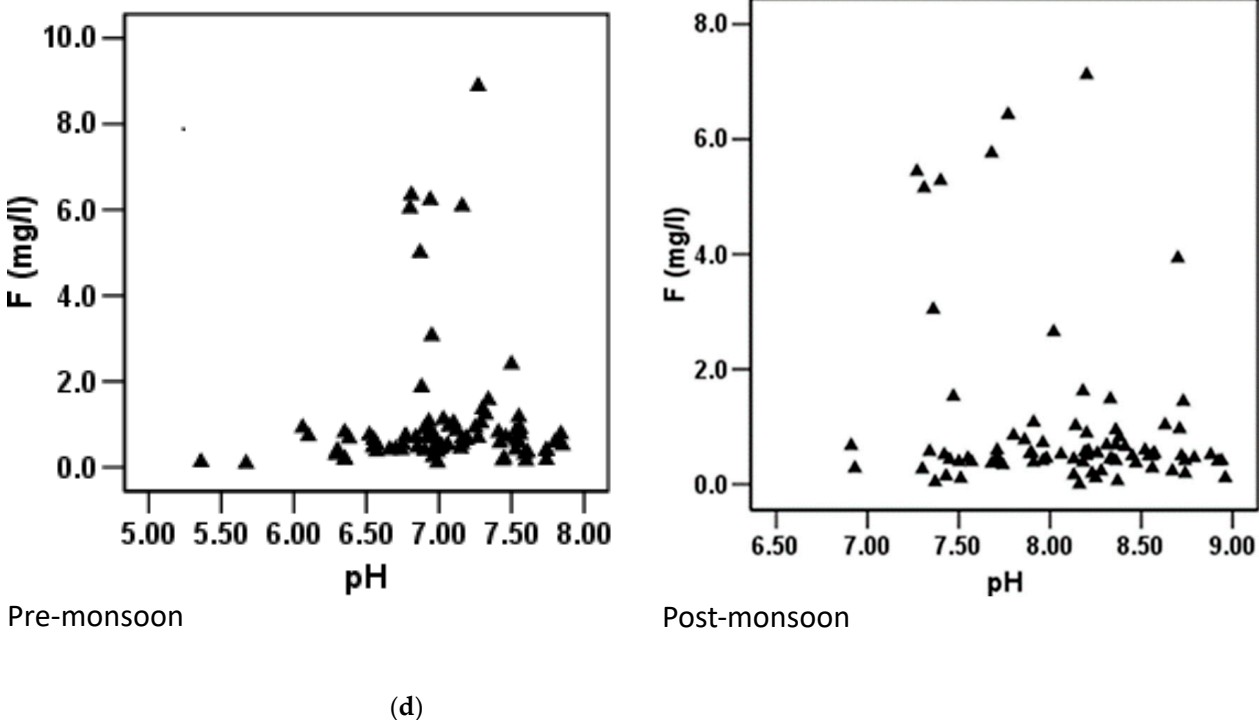

Pre-monsoon  Post-monsoon

(**d**)

**Figure 11.** Scatter plots for pre-monsoon and post-monsoon periods: (**a**) $F^-$ vs. $Ca^{2+}$; (**b**) $F^-$ vs. $Na^+$; (**c**) $F^-$ vs. $HCO_3$-; (**d**) $F^-$ vs. pH.

The study area comprised E–W-trending deep-seated major faults in feldspathic/ferruginous sandstones of the Kamthi and Raniganj Formations, which allowed groundwater to move through the coal-bearing Barakar sandstone and favored the fluoride-bearing aquifers to dissolve in groundwater. In order to compare surface and subsurface variations in mineralogy of the Barakar sandstone encountered in the eastern part of the study area where groundwater contained high concentration of $F^-$, XRD and optical microscopy analyses of rock samples from four locations, three samples from the surface, and one sample from the subsurface (from 135 m below the ground surface, collected while drilling of new bore well) were performed. The presence of appreciable amounts of white mica was seen during field campaigns in the outcrops (Figure 4a). XRD and optical microscopy indicated that quartz and K-feldspars constituted the main mineralogy of the sandstones. In addition to quartz (39–82%) and K-feldspars (9–40%), substantial amounts of white mica, biotite, and clay minerals (9–26%) were found in all the four samples. However, the subsurface rock sample exhibited a friable nature because it was found in a zone that remained in contact with water, and it showed a high degree of weathering of micas. The micas and clay minerals occurring in the feldspathic sandstones of the Barakar Formation may possibly be an important source for releasing $F^-$ in groundwater under a favorable geochemical environment. This is because these minerals can contain high amounts of $F^-$ as a replacement for $OH^-$ [45]. Further, thick aquitard horizons of shale/clay/coal beds, containing micas and clay minerals, may have also played a role in releasing $F^-$ to groundwater via anion exchange due to water–rock interaction because the wells were not completely cased.

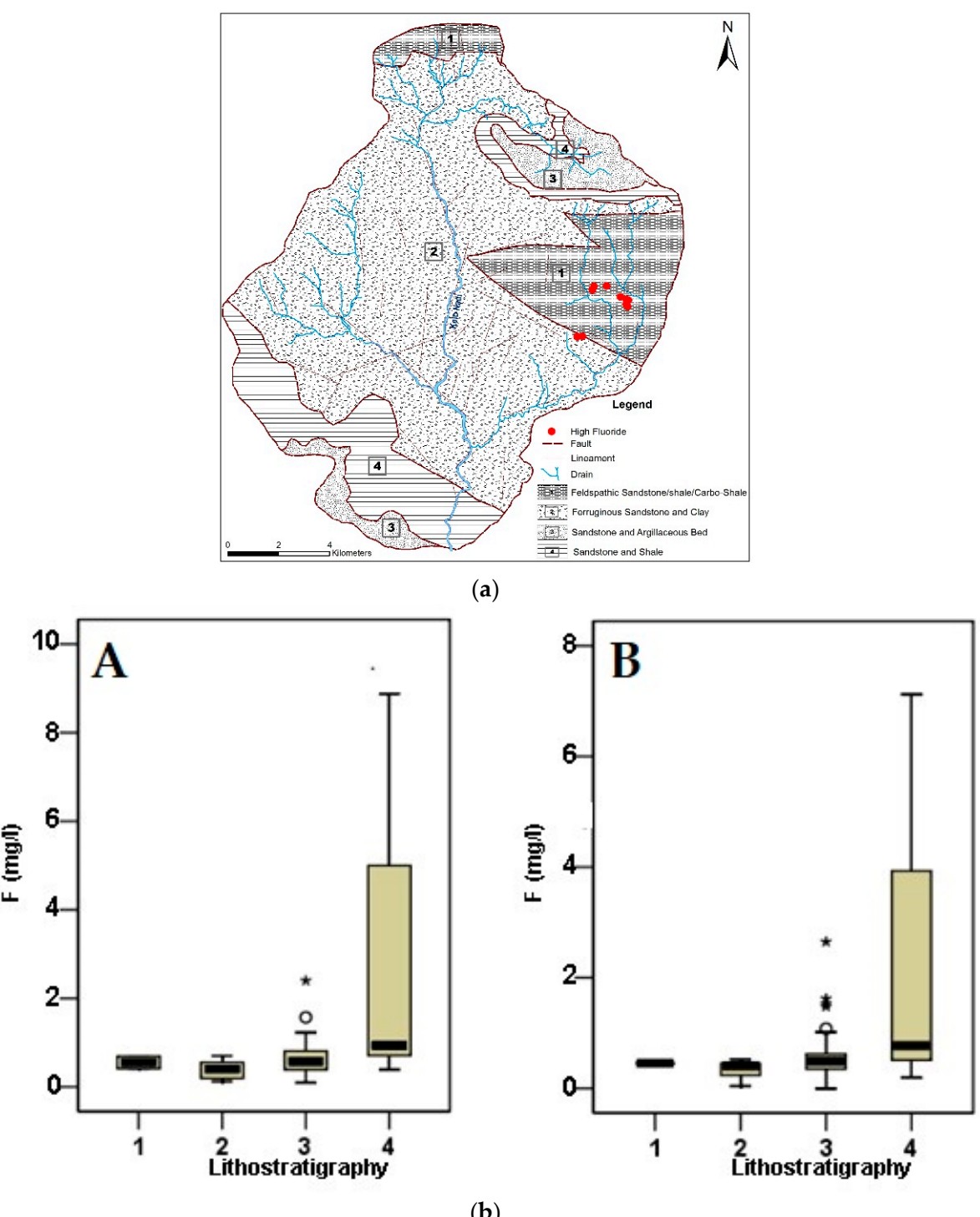

**Figure 12.** (**a**) Geological map of the study area showing locations of high-fluoride wells. (**b**) Box-and-whisker plots showing concentration of F⁻ with respect to lithostratigraphy during (**A**) pre-monsoon and (**B**) post-monsoon periods. 1 = Kamthi Formation, 2 = Raniganj Formation, 3 = Barren Measures Formation, 4 = Barakar Formation. Bars in the boxes indicate medians, open circles represent outliers, and stars represent extreme outliers.

### 3.8. Relationship of F⁻ with Well Depth

The scatter plot between F⁻ and well depth (Figure 13) showed a positive correlation, both for pre-monsoon (r = 0.616) and for post-monsoon (r = 0.707) periods. It is also clear from the scatter plot that a high F⁻ concentration was associated with wells in a depth range

of 110 to 150 m. The high F⁻ concentration in deeper wells may be explained by the intense dissolution of fluoride-bearing minerals present in rock formations due to a rise in the temperature and residence time of groundwater with gradually increasing depth [26,61]. In contrast, the low F⁻ concentration in groundwater from shallow aquifers/wells may be due to annual recharge from monsoon rainfall and lower temperature. Many researchers have reported that well depth is a significant parameter for enhancing fluoride concentration in groundwater [62–64]. The contact period of groundwater with fluoride-bearing minerals can also produce high fluoride content [25,62,65,66]. The high fluoride concentration observed in one subsurface rock sample collected from the area during drilling could also have been due to the fact that drilling was performed in this area at substantial depth and intersected the strata containing fluoride-bearing minerals with super-saturated biotite due to the weathering effect underlying the Barakar (coal-bearing) sandstone, further releasing F⁻ in higher concentrations [67].

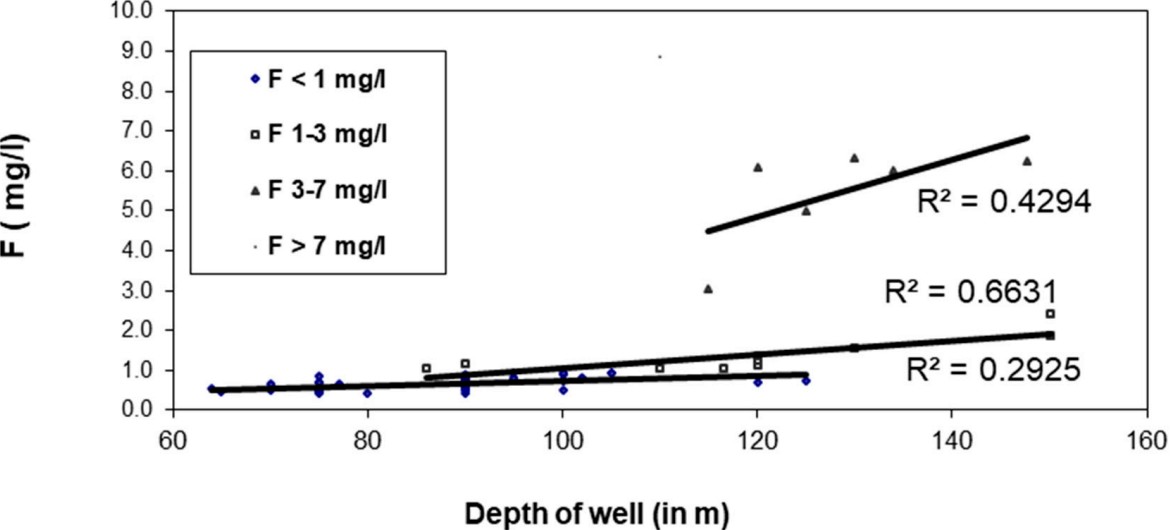

**Figure 13.** Scatter plot of F⁻ content in groundwater versus well depth.

The fluoride concentration was high in zones B and C. Thus, in order to compare the high F⁻ content and groundwater depth (F⁻ > 1.2 mg/L), the groundwater samples of the study area located in zones B and C were grouped into three categories i.e., pre-monsoon, mid-monsoon, and post-monsoon periods, with respect to the depth of the bore wells. The depth range of wells varied between 70 m and 150 m bgl. Hydrochemical properties varied in all three seasons (Table 6). In the pre-monsoon period, the pH range was 6.55–7.84, while, in the mid-monsoon period, it ranged from 6.22 to 7.64, and, in the post-monsoon period, it ranged from 7.92 to 8.94. The dominant major $Na^+$ ion concentration varied over the ranges of 6.02–62.65 mg/L in the pre-monsoon period, 2.71–50.41 mg/L in the mid-monsoon period, and 2.12–53.19 mg/L in the post-monsoon period, while $Ca^{2+}$, $Mg^{2+}$, and $K^+$ ion concentrations were relatively stable in all the three seasons. The concentration of $HCO_3^-$ was almost stable in all three seasons with a pre-monsoon range of 77–386 mg/L, mid-monsoon range of 79.3–414.8 mg/L, and post-monsoon range of 72.5–404.7 mg/L. $SO_4^{2-}$, $Cl^-$, and $NO_3^-$ ion concentrations remained the same in all three periods except in one well during the post-monsoon season (well No. 52), where the $SO_4^{2-}$ concentration was observed at 215 mg/L at greater depth. A gradual increase in F⁻ ion concentration of 0.41–8.88 mg/L, 0.8–7.12 mg/L, and 0.14–7.15 mg/L in the pre-, mid-, and post-monsoon periods was observed as the depth range of wells increased from 70 to 150 m bgl. F⁻ (i.e., >1.2 mg/L) occurred mainly in mixed types of water (Na-Ca-HCO₃, Na-Ca-Mg-HCO₃, and Na-Mg-Ca-HCO₃), where the $Na^+$ concentration was relatively higher than that of other cations compared to water of the Ca-Mg-HCO₃ type (Figure 8).

**Table 6.** Physicochemical properties of water samples (zones B and C) of three seasons (pre-, mid-, and post-monsoon periods) (units, mg/L), except for the well depth and pH.

| Water Type | Well Depth (m) | pH | EC | TDS | TH | ALK | $Ca^{2+}$ | $Mg^{2+}$ | $Na^+$ | $K^+$ | $NO_3^-$ | $HCO_3^-$ | $Cl^-$ | $SO_4^{2-}$ | $F^-$ |
|---|---|---|---|---|---|---|---|---|---|---|---|---|---|---|---|
| **Pre-monsoon period** | MAX | | | | | | | | | | | | | | |
| | 150 | 7.84 | 788 | 433 | 257.4 | 316.78 | 49.5 | 32.51 | 62.65 | 36.02 | 2.89 | 386 | 69.08 | 31.8 | 8.88 |
| | MIN 70 | 6.55 | 157.2 | 102 | 69 | 54.95 | 13.86 | 4.82 | 6.02 | 3.29 | 0.154 | 77 | 0 | 2.31 | 0.41 |
| | MEAN 112.58 | 7.19 | 387.36 | 244.45 | 126.61 | 152.08 | 25.31 | 15.29 | 27.00 | 17.38 | 0.84 | 186.48 | 15.12 | 12.73 | 2.69 |
| | MEDIAN 120 | 7.21 | 358.5 | 225 | 115.92 | 149.89 | 21.78 | 13.29 | 25.90 | 15.46 | 0.489 | 187.8 | 11.835 | 11.30 | 1.46 |
| | SD 24.66 | 0.31 | 137.74 | 80.84 | 51.01 | 61.29 | 9.20 | 7.35 | 13.76 | 10.18 | 0.76 | 73.92 | 14.60 | 6.89 | 2.65 |

| Water Type | Well Depth (m) | pH | EC | TDS | TH | ALK | $Ca^{2+}$ | $Mg^{2+}$ | $Na^+$ | $K^+$ | $NO_3^-$ | $HCO_3^-$ | $Cl^-$ | $SO_4^{2-}$ | $F^-$ |
|---|---|---|---|---|---|---|---|---|---|---|---|---|---|---|---|
| **Mid-monsoon period** | MAX 150 | 7.64 | 818 | 408 | 245 | 340 | 40.96 | 22.63 | 50.41 | 91.7 | 0.34 | 414.8 | 56.72 | 40.73 | 7.15 |
| | MIN 70 | 6.22 | 198 | 99 | 55 | 65 | 12.3 | 4.46 | 2.71 | 2.77 | 0.05 | 79.3 | 4.32 | 0.07 | 0.28 |
| | MEAN 112.58 | 6.99 | 435.85 | 217.9 | 133.25 | 166.5 | 22.24 | 10.58 | 21.59 | 15.65 | 0.19 | 203.13 | 16.99 | 8.05 | 2.56 |
| | MEDIAN 120 | 6.985 | 406 | 202.5 | 122.5 | 150 | 17.09 | 8.65 | 20.43 | 10.24 | 0.17 | 183 | 13.91 | 1.96 | 1.4 |
| | SD 24.66 | 0.32 | 178.06 | 88.84 | 57.31 | 67.61 | 9.15 | 5.48 | 10.91 | 19.40 | 0.10 | 82.49 | 12.55 | 12.80 | 2.25 |

| Water Type | Well Depth (m) | pH | EC | TDS | TH | ALK | $Ca^{2+}$ | $Mg^{2+}$ | $Na^+$ | $K^+$ | $NO_3^-$ | $HCO_3^-$ | $Cl^-$ | $SO_4^{2-}$ | $F^-$ |
|---|---|---|---|---|---|---|---|---|---|---|---|---|---|---|---|
| **Post-monsoon period** | MAX 150 | 8.92 | 1083 | 542 | 401 | 361.4 | 62.68 | 41.57 | 53.19 | 26.12 | 11.53 | 404.7 | 32.11 | 215 | 7.12 |
| | MIN 70 | 7.27 | 167 | 84 | 59.4 | 59.4 | 11 | 3.3 | 2.12 | 2.7 | 0 | 72.5 | 3.59 | 0.26 | 0.14 |
| | MEAN 112.58 | 7.98 | 445.2 | 222.75 | 151.48 | 150.32 | 22.96 | 12.74 | 21.26 | 11.76 | 1.32 | 177.99 | 13.01 | 15.83 | 2.72 |
| | MEDIAN 120 | 7.91 | 357.5 | 178.5 | 112.65 | 126.25 | 16.25 | 8.62 | 21.16 | 9.12 | 0.53 | 154 | 11.33 | 4.32 | 1.57 |
| | SD 24.66 | 0.54 | 243.35 | 121.71 | 95.69 | 87.51 | 13.97 | 10.19 | 12.73 | 7.51 | 2.49 | 97.20 | 7.95 | 47.35 | 2.34 |

## 4. Discussion

The results of the hydro-geochemical evaluation of groundwater, highlighted in the previous section, can be used to constrain the origin source and hydro-geochemical processes. Geographical and geological locations were major elements catalyzing the enrichment of $F^-$ concentration in bore wells located in the study area. The main cations of groundwater were $Ca^{2+}$, $Mg^{2+}$, $Na^+$, and $K^+$, and the main anions were $HCO3^-$, $SO42^-$, $Cl^-$, and $NO3^-$. The groundwater chemistry of the micro-watershed changed in all three major flow directions, i.e., west to east, zone A, north to east, zone B, and east to southeast, zone C (Figure 2). When comparing all three zones, it was observed that the spatial distribution of $F^-$ (Figure 7a,b) in the groundwater of the study area showed that bore wells with high $F^-$ concentrations were confined to eastern part (zone C) and increased along the flow path within the zone itself. The present study shows that high $F^-$ concentrations (>1.2 mg/L) in groundwater occurred in Muragaon, Saraitola, Pata, Kunjhemura, and Dolnara villages. The $F^-$ concentration was more or less independent of other water-soluble components; however, remarkably, no significant correlation existed between $F^-$ and pH. Fluoride solubility is lowest in low pH (5–6.5) [68], while ionic exchange takes place between $F^-$ and $OH^-$ ions at higher pH (illite, mica), consequently increasing the $F^-$ concentrations in groundwater [33,69]. The groundwater in this area was more or less alkaline with pH varying from 6.91 to 8.96; its pH mean value was 8.16 in the pre-monsoon period, indicates the alkaline characteristics of groundwater (Table 2). Groundwater with a high pH value favors the enrichment of $F^-$ [16,28–30] because fluoride ($F^-$) and hydroxyl ($OH^-$) ions have similar ionic radii, and hydroxyl ions in groundwater can displace exchangeable fluoride ions from fluoride-bearing minerals when alkaline groundwater circulates through the aquifer [45]. $F^-$ concentration is positively correlated with $Na+$ and pH, but negatively correlated with the $Ca^{2+}$, indicating that, in the aquifer, a high $F^-$ concentration is due to the involvement of geochemical processes in increasing $Na^+$ and pH and decreasing $Ca^{2+}$. Therefore, the geochemical parameters $Na^+$, pH, and $Ca^{2+}$ can explain the geochemical processes that might have been responsible for high $F^-$ in the groundwater of eastern part of the study area. On the other hand, groundwater types are not related to geology, whereas the source of $F^-$ might be related to geology. The geochemical behavior of groundwater is controlled by the geochemistry of groundwater.

The study area was sedimentary-dominant aquifer; dissolution of $F^-$ was a plausible cause for occurrence of $F^-$ concentrationin groundwater. An increase in $F^-$ concentration in groundwater was noticed as the $Na^+$ content increases (Figure 9). The plots of $F^-$ and lithogenic $Na^+$ indicated a remarkable positive correlation between $F^-$ and lithogenic $Na^+$ (Table 4 and Figure 11b). Piper's trilinear diagrams also showed $Na^+$ as the dominant cation, wherein the concentration of $F^-$ was high. A similar research finding was proposed by [31], who also reported that an increase in $F^-$ content in groundwater was linked to the geochemical processes corresponding to an increase in $Na^+$ concentration and decrease in $Ca^{2+}$ concentration. The $F^-$ concentration in groundwater is negatively correlated with calcium ions ($Ca^{2+}$) in groundwater. Such a relationship between $Ca^{2+}$ and $F^-$ was also found in our research, which was supported by the significant negative correlation ($-0.22$ in pre-monsoon period; $-0.33$ in post monsoon period). Thus, due to the dominance of ion-exchange processes functioning in the studied aquifer, an increasing $F^-$ content was shown to correlate with an increasing $Na^+$ content and decreasing $Ca^{2+}$. The $Ca^{2+}$ and $Na^+$ ion concentrations in aquifer increased from the recharge to discharge zone and flowed through the drainage direction (Figure 2). This interpretation was also supported by the Na/Ca ratio, which was three times greater for groundwater samples with $F^- > 1.2$ mg/L as compared to those with $F^- \leq 1.2$ mg/L (i.e., 0.42 vs. 1.16 in the pre-monsoon period and 0.46 vs. 1.32 in the post-monsoon period). This was due to ion exchange, whereby calcium ions in water may react with clay minerals to release $Na^+$ ions, thus increasing their concentration in groundwater [70]. A strong correlation between high $F^-$ and low $Ca^{2+}$ content in alkaline groundwater has also been reported by many authors (e.g., [31]),

wherein an increase in solubility of fluorine-bearing minerals with an increase in $Na^+$ concentration was observed [33].

Through the saturation index analysis, it was found that all groundwater samples were undersaturated with calcite, fluorite, halite, gypsum, anhydrite, and dolomite during the pre-monsoon period. This means that these minerals could dissolve more in groundwater, which could lead to an increase in their concentration. However, during the post-monsoon period, calcite and dolomite were found to be oversaturated, which means that no further dissolution could occur, and they would precipitate as $CaF_2$. On the other hand, fluorite remained undersaturated due to the oversaturation of calcite, which reduced calcium activity and allowed more fluorite to dissolve, thereby increasing the $F^-/Ca2+$ of the solution. This finding is consistent with the study conducted by Alamry [50].

The anthropogenic origin of high $F^-$ in groundwater in the study area could be completely ruled out. There was no such activity in the study area that could be considered as potential source of fluoride inputs in groundwater as contaminant. Thus, the high concentration of fluoride in groundwater in the study area was geogenic in origin i.e., hydro-geo-chemical conditions and coal-bearing formations were responsible for the higher concentration of $F^-$ in the bore wells in the study area. However, the process of $F^-$ enrichment is still not well understood [46,71]; many authors have accepted the general principle of exchangeable fluoride ($F^-$) ions by hydroxyl ($OH^-$) ions (e.g., [67,72]). The geochemistry of groundwater considers factors such as adsorption, dissolution, hydrolysis, precipitation, ion-exchange, and geo-chemical processes as principle reasons that contribute to the enrichment of $F^-$ concentration in groundwater [71]. For example, sodium bicarbonate-rich groundwater ($NaHCO_3$) in weathered rock formation accelerates the rate of dissolution of fluorite ($CaF_2$) to release $F^-$ into groundwater due to water–rock interaction with time, as shown in mass balance Equation (1). The presence of high bicarbonate ions ($HCO_3^-$), sodium ions ($Na^+$), and pH favors the release of $F^-$ into groundwater [33,73,74], thus mobilizing $F^-$ from fluorite ($CaF_2$) mineral (Equations (1) and (2));

$$CaF_2 + 2NaHCO_3 = CaCO_3 + 2Na^+ + 2F^- + H_2O + CO_2 \tag{1}$$

$$CaF_2 + 2HCO_3 = CaCO_3 + 2F^- + H_2O + CO_2 \tag{2}$$

Groundwater with high $HCO_3^-$ and $Na^+$ content is usually alkaline in nature [33] and has a relatively high $OH^-$ concentration (Equation (3)). $F^-$-rich minerals such as muscovite, biotite, and amphiboles have been reported in the study area, and fluoride ($F^-$) ion can replace hydroxyl ($OH^-$) ions under alkaline conditions. The reaction process of the replacement of hydroxyl ($OH^-$) ions by fluoride ions ($F^-$) from muscovite mineral [33] is illustrated below (Equation (4)).

$$HCO_3^- + H_2O = H_2CO_3 + OH^- \tag{3}$$

$$KAl_2 [AlSi_3O_{10}] F_2 + 2OH^- = KAl_2 [AlSi_3O_{10}] [2OH]_2 + 2F^- \tag{4}$$

The concentration of $F^-$ in the bore wells located in the study area largely depended upon factors like climate, evaporation, geology and precipitation. Such factors were also reported by Li et al. [60]. It is widely accepted by many researchers that the enrichment of $F^-$ in groundwater is due to persistent water–rock interaction [51,52,74,75].

$F^-$ concentration in the sedimentary aquifer of the study area had an apparent variation with well depth. The maximum F concentration of deep groundwater was 8.88 mg/L observed at the depth of 150 m, whereas F concentration was 0.41 mg/L at the depth of 70 m. Most high-F wells were observed in deep groundwater. The genesis of high F concentration may be attributed to ionic activity in deep groundwater that promotes water interaction and residence time over less deep groundwater.

Many researchers have reported that $F^-$ occurs in alkaline environments with higher $HCO_3^-$ [76–78] and is positively correlated with $HCO_3^-$ [33]. There were also instances where negative correlations between $F^-$ and $HCO_3^-$ have been reported in deeper aquifers [56,57]. However, in this particular study, there was a negative correlation between $F^-$ and $HCO_3^-$, in line with other studies that reported a negative correlation of $F^-$ in deep-seated wells similar to our conditions. The scatter plots of $F^-$ and $K^+$ showed a negative correlation, which may have been due to weathering of K-bearing minerals and/or the fixation of $K^+$ ions in micas and clay minerals [34]. It is argued by many researchers that micas and clay, which often occur as pertinent minerals in rocks and contain $F^-$ at the $OH^-$ sites, can form the dominant source for high $F^-$ incidence in groundwater via the process of anion exchange ($OH^-$ for $F^-$) takes place especially in sedimentary terrains [31,33,35]. The micas and clay minerals occurring abundantly in the lithological assemblage of the Barakar Formation could be considered the source of elevated $F^-$ concentration in groundwater. The presence of $Li^+$ (although occurring in very small concentrations and having a poor positive correlation with $F^-$) in the high-$F^-$ zone lends support to weathering of micas as the source of $F^-$ at a deeper level [44]. The absence of $PO_4^{3-}$ in bore wells samples rules out fluoride contribution from phosphatic minerals in data from all three seasons, along with the application of phosphatic fertilizers. The absence of industrial activities and association of the high $F^-$ zone with deeper aquifers further negate anthropogenic contamination. Because the wells are not completely cased, the collected groundwater samples represent a multiple-aquifer/aquitard system. Therefore, it is recommended to carry out depth-wise sampling of groundwater and rocks for all three seasons, which could not be conducted in the present study, to identify the subsurface horizon(s) and the constituent minerals causing the $F^-$ problem. Hydrogeochemical investigations are needed in the adjoining areas with similar geologic setting, particularly for the aquifers in the Barakar Formation, in order to delineate unsafe zones and take mitigation measures so as to safeguard people against the danger of consuming $F^-$-contaminated groundwater.

## 5. Conclusions

The study helped to gain insight to the source and hydro-geochemical processes that catalyze the high $F^-$ concentration in groundwater in a coal-bearing sedimentary (Gondwana) formation of Central India. The analysis of hydrochemical and geological datasets led to the following findings: (i) groundwater with high $F^-$ concentration occurred in the Barakar Formation, which has a litho-assemblage of feldspathic sandstones, shales, and coal; (ii) high $F^-$ concentration was mainly associated with Na-Ca-$HCO_3$, Na-Ca-Mg-$HCO_3$, and Na-Mg-Ca-$HCO_3$ types of groundwater; (iii) the $F^-$ concentration increased as the ratio of $Na^+$ and $Ca^{2+}$ increased ($Na^+$:$Ca^{2+}$, concentration in meq/L); (iv) $F^-$ had a significant positive correlation with $Na^+$ and $SiO_2$, and a significant negative correlation with $Ca^{2+}$, $Mg^{2+}$, $HCO_3^-$, and TH; (v) high $F^-$ concentration in groundwater was found in deeper wells. Micas and clay minerals, occurring in the feldspathic sandstones and intercalated shale/clay/coal beds, possibly formed an additional source for releasing $F^-$ in groundwater. Cation ($Ca^{2+}$ for $Na^+$) exchange appeared to be the dominant hydrogeochemical process operating in the study area. The incidence of high $F^-$ concentration in deeper wells indicated increased dissolution of mica because of higher residence time and temperature of groundwater in deeper aquifers, and consequently enhanced water–rock interaction. Climatic conditions such as semi-aridity and high temperature favor effective chemical weathering of these rocks. The absence of $PO_4^-$ in groundwater and industrial activities rules out the possibility of anthropogenic contamination. The results obtained in the present study can promote new research in this area for taking up a systematic hydrogeochemical investigation, and this is recommended in areas underlain by the Gondwana rocks, especially the coal-bearing Barakar Formation, where large populations may be at potential risk. This is especially pertinent because the area has great potential for mining and industrial development. People living in this area belong to tribal and schedule caste communities where groundwater hand-pumps are the only source of drinking water

supply for their daily requirements. Because people are uneducated, they have minimum awareness about the quality of water; hence, this is the main cause of prevailing dental and skeletal fluorosis. The health risk maps were generated in order to locate spatially the hotspot area areas at high risk due to high fluoride concentration. The government should act to develop the surface water supply (rainwater harvesting for the resident in the hotspot areas). In addition, the groundwater quality should be improved by installing the water purification plants in the hotspot villages at risk to safeguard the health of the residents.

Many initiatives have been undertaken by Rajiv Gandhi National Drinking Water Mission (RGNDWM) and the National Jaljeevan Mission (NJM) Government of India, Ministry of Jalshakti, Department of Drinking Water and Sanitation. In the area, the Public Health Engineering Department (PHED) has to create a public awareness on a large scale about fluoride issues and provide fluoride-free water to the villages. However, in rural India, villagers depend on hand-pumps for the drinking water supply; these hand-pumps are not systematically monitored, and their data are not properly recorded. Thus, the majority of the population living in rural area is still at risk of fluorosis. Therefore, the present research can motivate the government to analyze and monitor water quality in a scientific way twice a year (pre- and post-monsoon periods) so as to provide safe drinking water to human population residing in rural areas.

**Author Contributions:** Conceptualization, M.K.B., N.K., S.K.S. and E.J.M.C.; methodology, M.K.B., N.K., S.K.S. and E.J.M.C.; software, M.K.B.; validation, M.K.B.; formal analysis, M.K.B., N.K., S.K.S. and E.J.M.C.; investigation, M.K.B., N.K., S.K.S. and E.J.M.C.; resources, M.K.B.; data curation, M.K.B.; writing—original draft preparation, M.K.B.; writing—review and editing, M.K.B., N.K., S.K.S. and E.J.M.C.; visualization, M.K.B.; project administration, M.K.B.; funding acquisition, M.K.B. All authors have read and agreed to the published version of the manuscript.

**Funding:** This research received no external funding.

**Data Availability Statement:** Data will be made available on reasonable request.

**Acknowledgments:** The first author (M.K.B.) is thankful to V.K. Dadhwal, the then Dean, IIRS, for providing the necessary facilities and support to carry out this study. He is grateful to J.B. de Smith, ITC Netherlands, and Shri Mudit Kumar Singh, Chhattisgarh Council of Science and Technology, Raipur, for their support, guidance, and encouragement in writing the manuscript. G. Sakaram, National Geophysical Research Institute (NGRI), Hyderabad is extremely thanked for providing the support and guidance for saturation index concepts and analysis. K.S. Patel, Dhananjay Sahu, Gopal Krishan, Mahendra Singh, Amit Multania, and P.K. Mukherjee are thanked for their help in chemical/lab analysis. The officials of the Public Health Engineering Department (PHED) of Chhattisgarh, especially Hingorani and R.K. Tandan, are also thanked for their keen interest in this study and assistance during field campaigns. The Wadia Institute of Himalayan Geology, Dehradun, is thanked for XRD and microscopic analysis.

**Conflicts of Interest:** The authors declare no conflict of interest.

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
