# Peer review of "Interpretation of Fluoride Groundwater Contamination in Tamnar Area, Raigarh, Chhattisgarh, India"

_2673-4834, doi:10.3390/earth4030033_

Round 1

Reviewer 1 Report

It is a very good paper that provides a great insight in the relation between fluoride concentration in the groundwater and the geochemical background. Indeed, endemic fluorosis is a serious concern, therefore this paper it should be of high interest to readers and a starting point for future research. 

The paper may be published, subject to minor revision. 

Abstract - lines 12-13 Rephrase (endemic fluorosis is a disease - includes health issues) 

Lines 189-190 - Additional investigations involving SEM - EDS might be useful in defining a complete frame of the situation. 

References - More recent publications might be cited. 

-

Author Response

We would like to thank the editor and the reviewers for their time and effort in reviewing our paper and providing valuable and insightful comments, which helped us to improve the overall quality of our manuscript. We have addressed all the comments and suggestions in the revised manuscript and we hope that the current version meets your expectations. The whole manuscript has been thoroughly checked and English has been improved in throughout the text.

In the following, you can find a point-to-point response to all comments. Please see the detailed responses below.

Reviewer 1

It is a very good paper that provides a great insight in the relation between fluoride concentration in the groundwater and the geochemical background. Indeed, endemic fluorosis is a serious concern, therefore this paper it should be of high interest to readers and a starting point for future research.

The paper may be published, subject to minor revision.

Response: Thanks a lot for your motivating remarks. We carefully checked and made the minor corrections suggested by the reviewer.

Comment 1. Abstract - lines 12-13 Rephrase (endemic fluorosis is a disease - includes health issues)

Response: Thanks for pointing this. The paragraph is revised as: High concentration of fluoride (F¯) in drinking water is harmful and is a serious concern worldwide due to its toxicity and accumulation in the human body. There are various sources of fluoride (F¯) and divergent pathways to enter into groundwater source and poses adverse effect on human health.

Comment 2. Lines 189-190 - Additional investigations involving SEM - EDS might be useful in defining a complete frame of the situation.

Response: SEM & EDS gives an idea of elemental composition analysis. we have already done XRD analysis for Barakar sandstone (main aquifer lithology) to identify mineral composition.  

Comment 3. References - More recent publications might be cited.

Response: OK, as suggested some latest references have been added.

Reviewer 2 Report

1. Abstract need to revise and add the key findings of the present study 

2. Fig.1 Sample locations details were missing 

3. Merge Fig.1 and 3, use different color or symbol for varying monsoon season 

4. Based on the results, Nitrate concentration were high but the study only focused on F-. Why?

5. Categorize the piper trilinear diagram and highlight the different class in Fig.6

6. Fig.10 is not clear , is it spatial diagram ? how much area was contaminated ?

7. Revise the conclusion based on the correction carried out in the main text 

Moderate correction required 

Author Response

We would like to thank the editor and the reviewers for their time and effort in reviewing our paper and providing valuable and insightful comments, which helped us to improve the overall quality of our manuscript. We have addressed all the comments and suggestions in the revised manuscript and we hope that the current version meets your expectations. The whole manuscript has been thoroughly checked and English has been improved in throughout the text.

In the following, you can find a point-to-point response to all comments. Please see the detailed responses below.

Reviewer 2

Comments and Suggestions for Authors

Comment 1. Abstract need to revise and add the key findings of the present study

Response: The abstract has been revised as suggested.

Comment 2. Fig.1 Sample locations details were missing

Response: Fig 1 shows the geological setup of the area while the sample locations are depicted in Fig. 3. However, as per the reviewer's comment, Fig 1& 3 have been merged and presented as Figure 1 in the revised version.

Comment 3. Merge Fig.1 and 3, use different color or symbol for varying monsoon season

Response: Figure 1 is improved with showing the spatial location of sampling points. However, we keep Figure 3 for better informing the sampling locations in pre and post-monsoon and during the monsoon season. Additionally, we added a new Figure 2 to show the groundwater flow direction.

Comment 4. Based on the results, Nitrate concentration were high but the study only focused on F-. Why?

Response: The study was focused on high F concentration in groundwater, particularly coal bearing sandstone region. No considerable relation between fluoride and NO3 exists. Nitrate (NO3-) content ranges from a negligible amount to about 106.3 mg/l; Moreover, only one sample at Amaght village exceeds the recommended limit of 45 mg/l. and the rest of the area it is negligible. Thus, we focused on F-, which is a major concern in the study area.

Comment 5. Categorize the piper trilinear diagram and highlight the different class in Fig.6

Response: OK, as suggested we categorize the piper trilinear diagram and highlighted the different class. Please check Figure 8.

Comment 6. Fig.10 is not clear, is it spatial diagram? how much area was contaminated?

Response: Fig 10 is a box and whiskar plot which is used for graphically demonstrating the skewness groups of numerical geochemical data. Box and whisker plots showing concentration of F¯ with respect to geology 1 = Kamthi Formation, 2 = Raniganj Formation, 3 = Barren Measure Formation, 4 = Barakar Formation. It depicts that the plot high fluoride wells are confined to Barakar formation only. 

Comments 7. Revise the conclusion based on the correction carried out in the main text

Response: Modifications have been made.

Reviewer 3 Report

REVIEW OF: earth-246181, Hydro-geochemical analysis of fluoride contamination in groundwater to understand fluoride release into drinking water sources of Tamnar area, Raigarh, Chhatisgarh, India.

Specific Comments

1.     I have highlighted in yellow, words that need correction. Several edits are on the reviewed manuscript.

2.     Title is too long, and a hyphen is not needed in hydrogeochemical nor should the word be broken into two words, it is one word. For the title, I suggest: “Interpretation of fluoride groundwater contamination in Tamnar area, Raigarh, Chhatisgarh, India.” This title uses less than half the number of words and contains the same information.

3.     The authors are unaware of two important publications:

a.      Nordstrom and Smedley (2022) Fluoride in Groundwater, The Groundwater Project, gw-project.

b.     Nordstrom (2022) Fluoride in thermal and non-thermal groundwater: Insights from geochemical modeling, STOTEN 824, 153606.

c.      Both of these papers detail the conditions giving rise to high F and compile reports from more than 80 countries having high F in groundwater. With these references there would be less need many of the other references that were cited.

4.     There are places where subscripts and superscripts were not used, e.g., lines 170-172.

5.     The description for how field measurements were obtained is inadequate. The authors state that standing water was pumped out of the well before sampling but fail to mention how that was done. A specified number of pore volumes? Fast or slow pumping to remove standing water? These procedures affect the water chemistry. How was the pH measured? Were standard buffers used? If so, which ones? Did they bracket the pH of the samples? Did the water contact the air before or during the pH measurement? If so, it might affect the pH measurement. Were standards used for EC? Were the samples filtered in the field? If not, why not? If they were, what was the filter pore size? Were the samples preserved at all or in any way? Were the samples kept cool on storage?

6.     Inconsistent use of chemical symbols is apparent. See, for example, page 7 where the authors jump back and forth between chemical symbols and spelled out names for dissolved constituents. I prefer using chemical symbols throughout unless there is a clear need to do otherwise.

7.     Figure 9 scatter plots are helpful, but the linear regressions done on them is not at all helpful.

8.     The unidentified antecedent problem occurs in several places in the text, for example, see page 18. “This” is a pronoun and qualifies a noun. When there is no noun in the sentence, you have an unidentified antecedent.

Overall Comments

This paper reports on sampled groundwater wells in the Raigarh District of Chhatisgarh State, India and found high fluoride in many of the wells. I have not heard or read about fluoride studies in this area so that is important. The study looked at pre-monsoon, during monsoon, and post-monsoon chemical compositions which was a very good idea. Unfortunately, there seems to be little difference in general but the results are only shown as averages which is not helpful. I would say that a better approach would be to look at the same individual well for the three (or more?) samplings during the year to see if there are any trends. What the authors have done is to hide potential information in summaries of means, medians, and standard deviations. Figure 11 is one of the more intriguing plots because it looks like individual well data for fluoride and well depth. However, forget the linear regressions and the R2 values. They add nothing. The obvious trends can be seen without them. There are important pieces of information missing from Figure 11. Where in the aquifer system are these wells. How are the fluoride concentrations changing with the potentiometric surface or the general direction of groundwater flow?

The authors need to identify upgradient and downgradient directions of the general flow path and then look for trends in the major ions to look for chemical evolution of the groundwater from recharge to discharge such as “freshening” which is the evolution from a Ca-HCO3 “hard” water to a Na-HCO3 “soft” water. This evolution typically drives pH values higher and F concentrations higher and it is usually an ion-exchange process (Ca replacing Na on exchange sites as the authors have noted) accompanied by calcite precipitation which also removes Ca. This process is described well in several papers and I refer the authors to Nordstrom and Smedley (2022) and Nordstrom (2022) and references therein for a recent synthesis.

The authors tried to identify F-bearing minerals in some rock samples and discussed the possibility of OH-F exchange by phyllosilicates (layer silicates). Again, the results were less than successful. However, for the range of concentrations seen in these groundwaters I would suggest it would take more than that process. No one yet has provided convincing proof that just by increasing the pH you can get mg/L increases in fluoride from phyllosilicate hydroxide lattice sites. It is more likely to be minute grains of either fluorite or a fluorapatite phase which would take a lot more effort to identify. The rest of the paper mostly re-iterates what many others have said in publications (especially from Indian authors) without offering anything new or innovative. The extensive statistical calculations do not really contribute much. There is some hope for this publication, but only after consideration of the comments made by this reviewer.

To summarize – the authors need to characterize the chemical evolution of the aquifer, look at chemical and well depth changes during different seasons for individual wells, enhance their sampling protocol description, and look at standard groundwater geochemistry parameters such as saturation indices. These changes might make a useful contribution. I would recommend publication only after serious major revision.

The English quality is not bad, but it could use a little polishing as per suggestions on manuscript and in review notes. Carefully read through the paper before resubmitting. 

Author Response

We would like to thank the editor and the reviewers for their time and effort in reviewing our paper and providing valuable and insightful comments, which helped us to improve the overall quality of our manuscript. We have addressed all the comments and suggestions in the revised manuscript and we hope that the current version meets your expectations. The whole manuscript has been thoroughly checked and English has been improved in throughout the text.

In the following, you can find a point-to-point response to all comments. Please see the detailed responses below.

Reviewer 4

Specific Comments

Comment 1.  I have highlighted in yellow, words that need correction. Several edits are on the reviewed manuscript.

Response: Thanks a lot for the very fruitful corrections. We have considered all of them in our revised version.

Comment 2. Title is too long, and a hyphen is not needed in hydrogeochemical nor should the word be broken into two words, it is one word. For the title, I suggest: “Interpretation of fluoride groundwater contamination in Tamnar area, Raigarh, Chhatisgarh, India.” This title uses less than half the number of words and contains the same information.

Response: Thanks a lot and we completely agree with your suggestion. We changed the title as you suggested and considered hydrogeochemical as a single word.

Comment 3. The authors are unaware of two important publications:

  1. Nordstrom and Smedley (2022) Fluoride in Groundwater, The Groundwater Project,gw-project.
  2. Nordstrom (2022) Fluoride in thermal and non-thermal groundwater: Insights from geochemical modeling, STOTEN 824, 153606.
  3. Both of these papers detail the conditions giving rise to high F and compile reports from more than 80 countries having high F in groundwater. With these references there would be less need many of the other references that were cited.

Response: Thank you very much for sharing the useful references. I have added them to our revised version and found the book on Fluoride to be very informative. Additionally, the video was impressive and we were highly impressed by the work of Nordstorm et al.

Comment 4.  There are places where subscripts and superscripts were not used, e.g., lines 170-172.

Response: Thank you! Corrections have been applied.

Comment 5. The description for how field measurements were obtained is inadequate. The authors state that standing water was pumped out of the well before sampling but fail to mention how that was done. A specified number of pore volumes? Fast or slow pumping to remove standing water? These procedures affect the water chemistry. How was the pH measured? Were standard buffers used? If so, which ones? Did they bracket the pH of the samples? Did the water contact the air before or during the pH measurement? If so, it might affect the pH measurement. Were standards used for EC? Were the samples filtered in the field? If not, why not? If they were, what was the filter pore size? Were the samples preserved at all or in any way? Were the samples kept cool on µm membrane filters storage?

Response: Thanks a lot!! As suggested we have improved the description and added details.

Comment 6.  Inconsistent use of chemical symbols is apparent. See, for example, page 7 where the authors jump back and forth between chemical symbols and spelled out names for dissolved constituents. I prefer using chemical symbols throughout unless there is a clear need to do otherwise.

Response: Thank you! Corrections have been applied as suggested.  

Comment 7. Figure 9 scatter plots are helpful, but the linear regressions done on them is not at all helpful.

Response: We agree with the reviewer. However, as it is mentioned in several scientific articles we would like to keep it as it is.

Comment 8.  The unidentified antecedent problem occurs in several places in the text, for example, see page 18. “This” is a pronoun and qualifies a noun. When there is no noun in the sentence, you have an unidentified antecedent.

Response: Thank you! We carefully checked and made the corrections throughout the text

Overall Comments

This paper reports on sampled groundwater wells in the Raigarh District of Chhatisgarh State, India and found high fluoride in many of the wells. I have not heard or read about fluoride studies in this area so that is important. The study looked at pre-monsoon, during monsoon, and post-monsoon chemical compositions which was a very good idea. Unfortunately, there seems to be little difference in general but the results are only shown as averages which is not helpful. I would say that a better approach would be to look at the same individual well for the three (or more?) samplings during the year to see if there are any trends. What the authors have done is to hide potential information in summaries of means, medians, and standard deviations. Figure 11 is one of the more intriguing plots because it looks like individual well data for fluoride and well depth. However, forget the linear regressions and the R2 values. They add nothing. The obvious trends can be seen without them. There are important pieces of information missing from Figure 11. Where in the aquifer system are these wells? How are the fluoride concentrations changing with the potentiometric surface or the general direction of groundwater flow?

The authors need to identify upgradient and downgradient directions of the general flow path and then look for trends in the major ions to look for chemical evolution of the groundwater from recharge to discharge such as “freshening” which is the evolution from a Ca-HCO3 “hard” water to a Na-HCO3 “soft” water. This evolution typically drives pH values higher and F concentrations higher and it is usually an ion-exchange process (Ca) replacing Na on exchange sites as the authors have noted) accompanied by calcite precipitation which also removes Ca. This process is described well in several papers and I refer the authors to Nordstrom and Smedley (2022) and Nordstrom (2022) and references therein for a recent synthesis.

The authors tried to identify F-bearing minerals in some rock samples and discussed the possibility of OH-F exchange by phyllosilicates (layer silicates). Again, the results were less than successful. However, for the range of concentrations seen in these groundwater, I would suggest it would take more than that process. No one yet has provided convincing proof that just by increasing the pH you can get mg/L increases in fluoride from phyllosilicate hydroxide lattice sites. It is more likely to be minute grains of either fluorite or a fluorapatite phase which would take a lot more effort to identify. The rest of the paper mostly re-iterates what many others have said in publications (especially from Indian authors) without offering anything new or innovative. The extensive statistical calculations do not really contribute much. There is some hope for this publication, but only after consideration of the comments made by this reviewer.

To summarize – the authors need to characterize the chemical evolution of the aquifer, look at chemical and well depth changes during different seasons for individual wells, enhance their sampling protocol description, and look at standard groundwater geochemistry parameters such as saturation Indices. These changes might make a useful contribution. I would recommend publication only after serious major revision.

Response: Dear Reviewer, we want to express our gratitude for your excellent feedback on our manuscript. Your in-depth understanding of the subject matter is greatly appreciated. We have taken into consideration almost all of the comments and suggestions you provided, as they were extremely helpful. We have made corrections throughout the text, which you can view in the track change mode version of the revised manuscript. Thank you again for taking the time to review our work.

Comment: The English quality is not bad, but it could use a little polishing as per suggestions on manuscript and in review notes. Carefully read through the paper before resubmitting.

Response: The whole manuscript has been thoroughly reviewed and English corrections have been applied throughout the text.

Reviewer 4 Report

This paper reported by Dr. Mirza Kaleem Beg et al. focuses on hydrogeochemical analysis of fluoride contamination in groundwater in Tamnar area of India. The topic is very interesting and within the scope of Journal Earth. However, there are some major weaknesses. First, authors declare that fluoride contamination in groundwater in the study area is geogenic. It is arbitrary, because this paper did not show evidences that human activities such as urbanization and industrialization had little influence on the occurrence of fluoride contamination in groundwater in the study area. In my opinion, authors need to supplement data related to human activities in the study area, and discuss the impact of human activities on fluoride contamination in groundwater in the study area. Second, more useful methods such as principal component analysis may be used in this study to further analyze hydrogeochemical processes controlling fluoride contamination in groundwater. In addition, the written is weak, and the English level should be improved greatly. Therefore, I recommend a major revision for this manuscript before acceptable.

In addition, I show some references related to principal component analysis for understanding hydrogeochemical processes that may be useful to authors.

A sharp contrasting occurrence of iron-rich groundwater in the Pearl River Delta during the past dozen years (2006-2018): the genesis and mitigation effect. Science of the Total Environment, 829, 154676, 2022

Geochemical factors controlling natural background levels of phosphate in various groundwater units in a large-scale urbanized area. Journal of Hydrology, 608, 127594, 2022

Spatial distribution and origin of shallow groundwater iodide in a rapidly urbanized delta: A case study of the Pearl River Delta. Journal of Hydrology 585, 124860, 2020

the written is weak, and the English level should be improved greatly. 

Author Response

We would like to thank the editor and the reviewers for their time and effort in reviewing our paper and providing valuable and insightful comments, which helped us to improve the overall quality of our manuscript. We have addressed all the comments and suggestions in the revised manuscript and we hope that the current version meets your expectations. The whole manuscript has been thoroughly checked and English has been improved in throughout the text.

In the following, you can find a point-to-point response to all comments. Please see the detailed responses below.

Reviewer 3

This paper reported by Dr. Mirza Kaleem Beg et al. focuses on hydrogeochemical analysis of fluoride contamination in groundwater in Tamnar area of India. The topic is very interesting and within the scope of Journal Earth. However, there are some major weaknesses. First, authors declare that fluoride contamination in groundwater in the study area is geogenic. It is arbitrary, because this paper did not show evidence that human activities such as urbanization and industrialization had little influence on the occurrence of fluoride contamination in groundwater in the study area. In my opinion, authors need to supplement data related to human activities in the study area, and discuss the impact of human activities on fluoride contamination in groundwater in the study area. Second, more useful methods such as principal component analysis may be used in this study to further analyze hydrogeochemical processes controlling fluoride contamination in groundwater. In addition, the written is weak, and the English level should be improved greatly. Therefore, I recommend a major revision for this manuscript before acceptable.

Response: Thanks a lot for pointing out the important issues.

In our introduction, we mentioned that "Given the large population at risk due to rapid population growth and increasing mining and industrialization". However, we would like to clarify that this was in reference to the broader district context, and not specific to our study area. We apologize for any confusion this may have caused and for not clarifying this in our manuscript.

The study area is located 100 kilometers away from the headquarters of Raigarh district. It is a rural area where no industries are present, a fact that has been confirmed by satellite images. Bore wells are the only source of drinking water in the area. Groundwater samples show that there is no presence of PO4-, and since there are no industrial activities in the rural area, rules out the possibility of anthropogenic contamination. In addition, we have determined through our specific case study that the predominant cause of fluoride contamination in groundwater is geogenic. This is further supported by the fact that other anthropogenic sources, such as phosphatic fertilizer and urban waste, are not present in our study area.

Regarding PCA we agree with you that PCA should be used when the variables are strongly correlated. However, If the relationship is weak between variables, PCA does not work well to reduce data. In our case, the correlation between the variables are weak as indicated in the correlation matrix. Thus, we have not applied the PCA.

The whole manuscript has been thoroughly reviewed and English corrections have been applied throughout the text.

Round 2

Reviewer 2 Report

Accept

spell check and minor language correction

Reviewer 4 Report

accept

need minor editing